# Sediment transport by Greenland's icebergs

Ethan Pierce[1,2] ✉, Irina Overeem [2,3] & Bent Hasholt [4]

Ice-rafted debris (IRD) from Greenland's tidewater glaciers provides key inputs to biogeochemical cycles, sequesters sediment in fjords, and leaves evidence of paleoclimate conditions. Previous work has shown that most IRD is entrained in basal ice, but studies have yet to translate this process knowledge into predictions of IRD export. Here, we combine field data and numerical models to quantify sediment transport in basal ice and estimate IRD fluxes. We present 210 samples of debris-rich icebergs, collected from three fjord systems, showing a long-tailed distribution of sediment concentrations between 0.1% and 45% by mass (3.48% median). Then, we develop a numerical process model of erosion and entrainment to predict the thickness of debris-rich ice layers and the IRD flux from each outlet. Across our selected fjords, we show a first-order relationship between a catchment's ice yield and sediment yield. By extrapolating this relationship across Greenland's marine-terminating outlets, we estimate that icebergs export 454 Mt a$^{-1}$ of IRD (with a 95% confidence interval from 292 to 716 Mt a$^{-1}$), representing up to one-third of Greenland's total sediment transport. Our results demonstrate the significant role of icebergs in Greenland's sediment budget, improving our understanding of how sediment export from glaciers and ice sheets will change under a warming climate.

In Greenland's fjords, we often find icebergs with striking layers of blackened, sediment-rich ice, up to several meters thick (Fig. 1). Originating inland, at the base of the ice sheet, this ice-rafted debris (IRD) is a key source of sediment to the coastal ocean[1–4]. Within the grounding zone of glacial outlets, sediment deposits contribute to shoal formation, inhibiting tidewater glacier retreat[5–8]. In the Southern Ocean, iceberg drift tracks feature heightened chlorophyll levels, as primary producers harness nutrients from the sediment that rains out into the water columns[9–11]. While upwelling of deep ocean water is the primary source of macronutrients to Greenland's coastal waters[12], direct inputs from icebergs are an important source of micronutrients, such as iron and manganese[13]. In marine sediment cores, IRD deposits help us infer the behavior of large ice sheets in the paleo-record[14–17]. Despite its importance to downstream marine processes, IRD is the least well-constrained component of Greenland's sediment budget[3].

Although IRD contributes sediment to the North Atlantic Ocean, Greenland's fjords are the primary sink for sediment sourced from beneath the ice sheet. In a recent study, Andresen et al. estimated that Greenland's fjords receive 1.324 ± 0.79 Gt of sediment per year from the continent's marine-terminating glaciers (appx. $4.9 \times 10^8$ m³)[18]. This study identified a relationship between accumulation rates in marine sediment cores and meltwater runoff from adjacent catchments, suggesting that subglacial sediment transport plays a key role in delivering material to Greenland's fjords. Previous work by Overeem et al. estimated the sediment load carried by meltwater, based on a scaling relationship derived from remote sensing and field measurements[3]. They reconstruct meltwater-driven suspended sediment load based on the choice of scaling variable: 1.28 ± 0.51 Gt per year if extrapolating based on modeled runoff, or 0.892 ± 0.374 Gt per year if extrapolating based on the erosion potential of glaciated catchments. While these studies use different methods and have large enough uncertainties that they could be reconciled into a consistent sediment budget, the contribution from IRD remains poorly constrained.

Early estimates of IRD flux ranged over a full order of magnitude[19]. As more quantitative data on Greenland's ice fluxes became available, previous work assumed all of Greenland's outlet glaciers have a debris-rich basal ice layer with uniform thickness (3 m) and high sediment concentration (20–50% by mass)[1–3]. But observations from glacial

[1]Thayer School of Engineering, Dartmouth College, Hanover, NH, USA. [2]Institute of Arctic and Alpine Research, University of Colorado Boulder, Boulder, CO, USA. [3]Department of Geological Sciences, University of Colorado Boulder, Boulder, CO, USA. [4]Department of Geosciences and Natural Resource Management, University of Copenhagen, Copenhagen, Denmark. ✉e-mail: ethan.g.pierce@dartmouth.edu

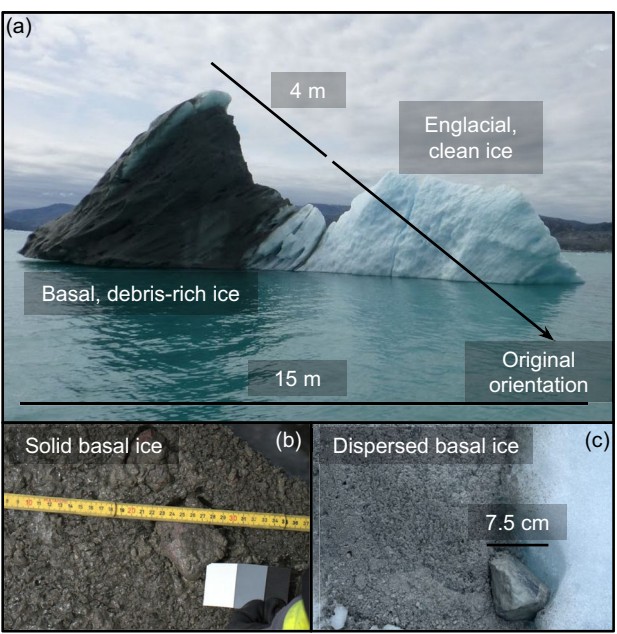

**Fig. 1 | Field photographs of ice-rafted debris.** The toppled iceberg that has exposed debris-rich basal ice layers (**a**). A solid basal ice facies, where ice has infiltrated the sediment matrix (**b**). A dispersed basal ice facies, where individual particles are emplaced within the ice matrix (**c**).

margins tell a much more complex story about the basal ice layer[20–22]. At the ice-bed interface, glaciers form a solid basal ice facies as ice infiltrates the underlying till[20]. This facies matches the prior assumption of sediment concentration in IRD layers, but often only extends for centimeters or decimeters into the ice[21]. Above it, we often observe several meters of dispersed basal ice, where individual particles are trapped within an intact ice matrix[22]. Dispersed basal ice can appear amber, brown, or even black in color, but only carries sediment concentrations of 1–10% by mass[21]. These distinctions matter: prior work has shown that a 3 m layer of 20% sediment would allow Greenland's icebergs to export 2.88 Gt a$^{-1}$ of IRD[3]. Not only is this estimate high compared to bulk sediment deposition in fjords[18], it would also comprise 22.5% of the total sediment flux from all rivers on Earth (12.8 Gt a$^{-1}$)[3]. To quantify Greenland's IRD export and better constrain its sediment budget, we develop a model that accounts for this variability in sediment entrainment processes.

In this work, we collect observations of 210 samples of ice-rafted debris entrained in 70 icebergs in three prominent fjord systems in both East and West Greenland. Guided by observations, we implement a process-based, numerical model of subglacial erosion, basal sediment entrainment, and advection by ice dynamics. Our model predicts sediment fluxes at an outlet glacier's terminus, representing the total source of sediment entering the fjord from ice discharge. From these results, we identify a scaling relationship between ice yield and IRD yield from a catchment. By extrapolating across the rest of the ice sheet, we estimate Greenland's icebergs deliver 454 Mt a$^{-1}$ (+262 Mt a$^{-1}$, −162 Mt a$^{-1}$) of sediment to its fjords per year.

## Results

### Field observations of ice-rafted debris
Between 2018 and 2022, we sampled sediment entrained in icebergs in three fjord systems: Ikerasak (west, near Disko Island, Fig. 2a), Kangertittivaq (east, also known as Scoresbysund, Fig. 2b), and Nuup Kangerlua (west, near Nuuk, Fig. 2c). In each fjord, we visually identified toppled icebergs with exposed debris-rich ice layers (Fig. 2e–f). These layers were often several meters thick and dark in color, offering a clean contact with the blue or white ice around them. The ice itself

had regular air bubbles, spaced centimeters to decimeters apart, elongated from shear. Sediment was entrained within the ice, either as individual particles, ranging from fine sands (0.1 mm) through cobbles (> 200 mm), or as aggregate clusters of fine silts (< 0.05 mm). We did not sample icebergs at random (c.f. ref. 4), but focused specifically on sampling exposed basal ice layers to inform a process-based modeling approach.

We took triplicate samples of each iceberg by hand. Then, we brought samples back to the laboratory, where we used a freeze-dryer to sublimate away the ice, and calculated rafted sediment concentrations (RSC) as the ratio of sediment mass to total mass in each sample. Across all three fjord systems, we find a long-tailed distribution in RSC, ranging from 0.1% to 45% by mass (Fig. 2d). The median RSC was 3.48%, with an interquartile range between 0.98% and 8.50%. True layer thicknesses were not possible to measure in the field, as we could rarely identify the original orientation of the iceberg. However, we did observe debris-rich layers with apparent thicknesses on the order of 1–5 meters in all three fjord systems. Despite not having access to the entire basal ice layer, our field measurements still clearly indicate that the sediment-rich ice beneath an outlet glacier is not uniform. Even within the same iceberg, sediment concentrations can vary by as much as 20% among samples taken ~1 m apart, if those samples come from different basal ice facies (see Fig. 1b, c)[21]. Because fluxes scale directly with concentration, we must be careful to account for this variability in models of ice-rafted debris production.

### Modeling subglacial sediment entrainment
Informed by observations of in-situ ice-rafted debris, we model subglacial sediment entrainment through two different processes. *Frozen fringe* forms when the effective pressure (the difference between ice pressure and water pressure beneath the glacier) is great enough to allow ice to infiltrate underlying sediment[23,24]. This process is driven by a rate balance between heave, set by effective pressure, and basal melt[25]. Frozen fringe presents as a sediment matrix with interstitial ice and, therefore, carries high sediment concentrations. It also serves as the mechanism for initially adding sediment particles at the base of the ice mass. Particles that have been entrained in the frozen fringe are not static, however. In the presence of temperature gradients, individual grains migrate via *regelation* through the ice matrix[26,27]. Regelation occurs when premelted films of water form around the outside of the grain, leading to a thermodynamic buoyancy at the scale of individual particles that drives motion along a background gradient[26]. This process gives rise to a basal ice layer with scattered sediment particles, and thus low sediment concentrations, matching descriptions of dispersed basal ice facies[28]. In combination, frozen fringe growth and regelation provide a system of basal ice formation where sediment concentrations can vary vertically within the glacier. Because frozen fringe occurs nearest to the bed, we expect dispersed ice to melt out last, explaining why it comprises the majority of our sampled observations.

We numerically model the sediment flux from each outlet, $Q_s$, by:

$$Q_s = (H_f C_f + H_d C_d) u_b W, \tag{1}$$

where $H_f$ and $H_d$ are the thickness of frozen fringe and dispersed basal ice, $C_f$ and $C_d$ are the sediment concentration in each layer, $u_b$ is the glacial sliding velocity, and $W$ is the width of the terminus. We model sliding velocity as the residual of observed surface velocity[29] and ice deformation velocity, described by a shallow ice approximation[30]. We ignore the effects of margin fluctuations, taking the terminus geometry at its present-day position. Sediment concentration in the frozen fringe is inherited from till porosity and saturation, following[25], while concentration in the dispersed layer is the median of our observations from dispersed basal ice carried by icebergs (Fig. 2d). The remaining two components of sediment flux are the thickness of frozen fringe and dispersed basal ice. We limit frozen fringe growth

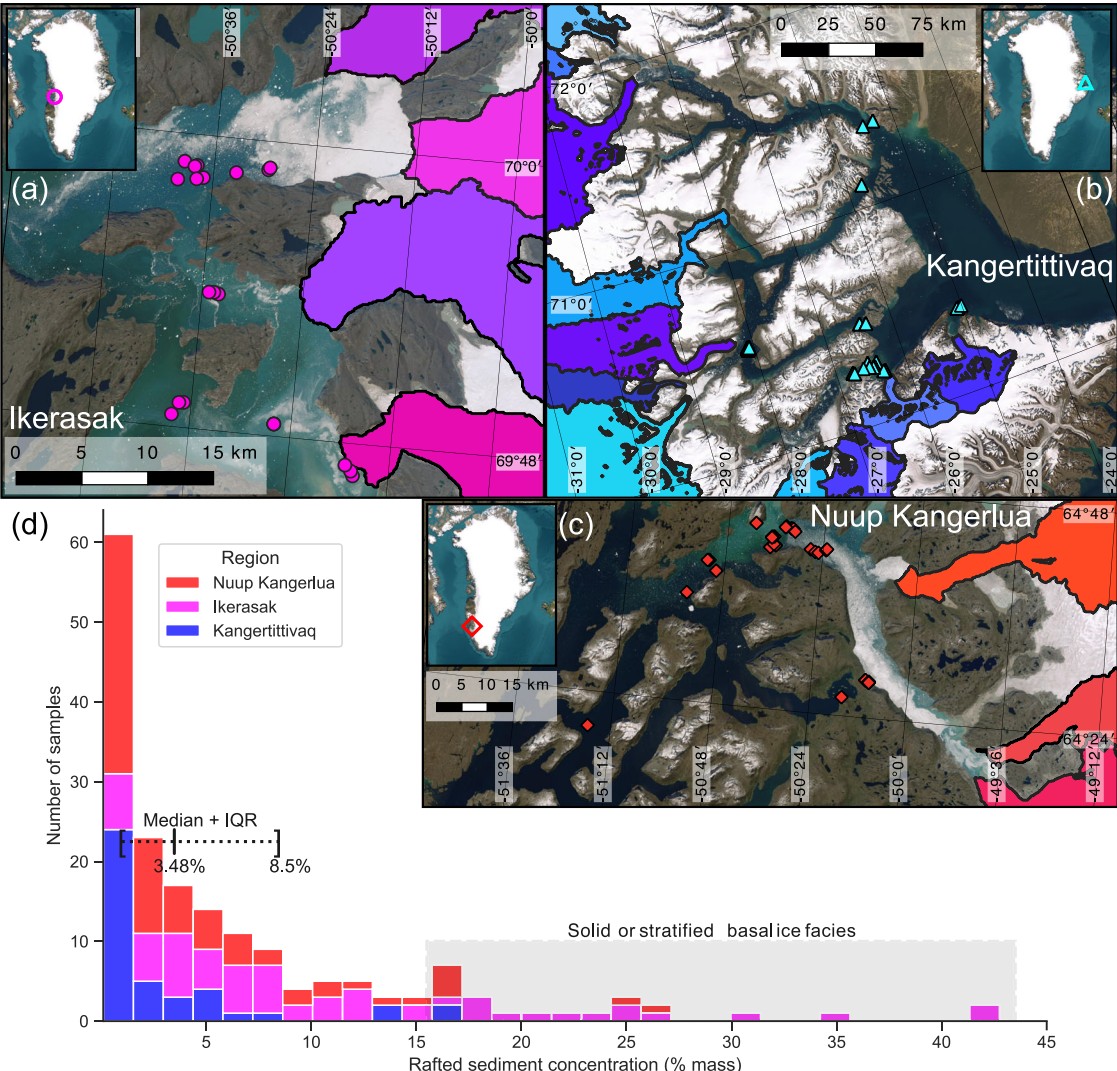

**Fig. 2 | Ice-rafted debris concentrations.** Maps of iceberg sample locations and marine-terminating outlet glacier catchments (**a**–**c**). Rafted sediment concentrations from debris-rich iceberg samples (**d**), with annotated median and interquartile range (IQR). Map image is the intellectual property of Esri and is used herein under license. Copyright ©2025 Esri and its licensors. All rights reserved.

based on the amount of eroded sediment available beneath the glacier, shutting off sediment entrainment completely where basal conditions are non-erosive. Similarly, dispersed layers only form where there is sediment already entrained in the underlying frozen fringe. Our numerical model solves two advection-reaction equations for the evolution of each of these basal ice layers. Each governing equation takes the form:

$$\frac{\partial h}{\partial t} + \nabla \cdot (u_b h) = S,$$  (2)

where $h$ is the layer thickness, $u_b$ is the sliding velocity, and $S$ is a source term that governs entrainment or deposition for each layer (see Methods). The lowermost layer, typically the frozen fringe, loses thickness from basal melt at each time step. We run the model until sediment fluxes at the catchment terminus achieve steady state (sediment entrainment increasing by < 0.1 mm a⁻¹). It takes each catchment less than 400 years for sediment fluxes at the terminus to reach a steady value, indicating that the modeled system reaches a dynamic equilibrium.

Our model results (Fig. 3) show that all of the outlet glaciers in our study areas entrain sediment and form a debris-rich basal ice layer.

Overall basal sediment fluxes in each fjord system are 24.58 Mt a⁻¹ in Nuup Kangerlua (southwest), 22.02 Mt a⁻¹ in Ikerasak (central-west), and 40.87 Mt a⁻¹ in Kangertittivaq (east). Frozen fringe thicknesses range from centimeters to a few meters, increasing towards the margin of the ice sheet (Fig. 3b). Towards the interior, we see patchy sediment entrainment, corresponding to regions with slow (< 1 m a⁻¹) and spatially variable sliding velocities. Dispersed layers have a similar thickness to the frozen fringe, but exhibit much less spatial variability, especially in the interior of the continent (Fig. 3a). At intermediate and lower elevations, sediment supply increases with basal velocity. The same advective transport that delivers ice to the calving front concentrates debris-rich ice, leading to the greatest frozen fringe and dispersed layer thicknesses, on the order of 1-3 meters. In the last several kilometers before the calving front, melt increases more rapidly than sediment entrainment. Most modeled glaciers carry a frozen fringe to the calving front, even while deposition from melt outpaces entrainment. At a few outlets, basal melt is potent enough to remove all of the frozen fringe, such that only the overlying dispersed ice survives to the calving front. Our model demonstrates that frozen fringe is a viable process for sediment entrainment across large areas of the ice sheet, and further that advection of basal ice plays a key role in concentrating entrained sediment towards the calving front.

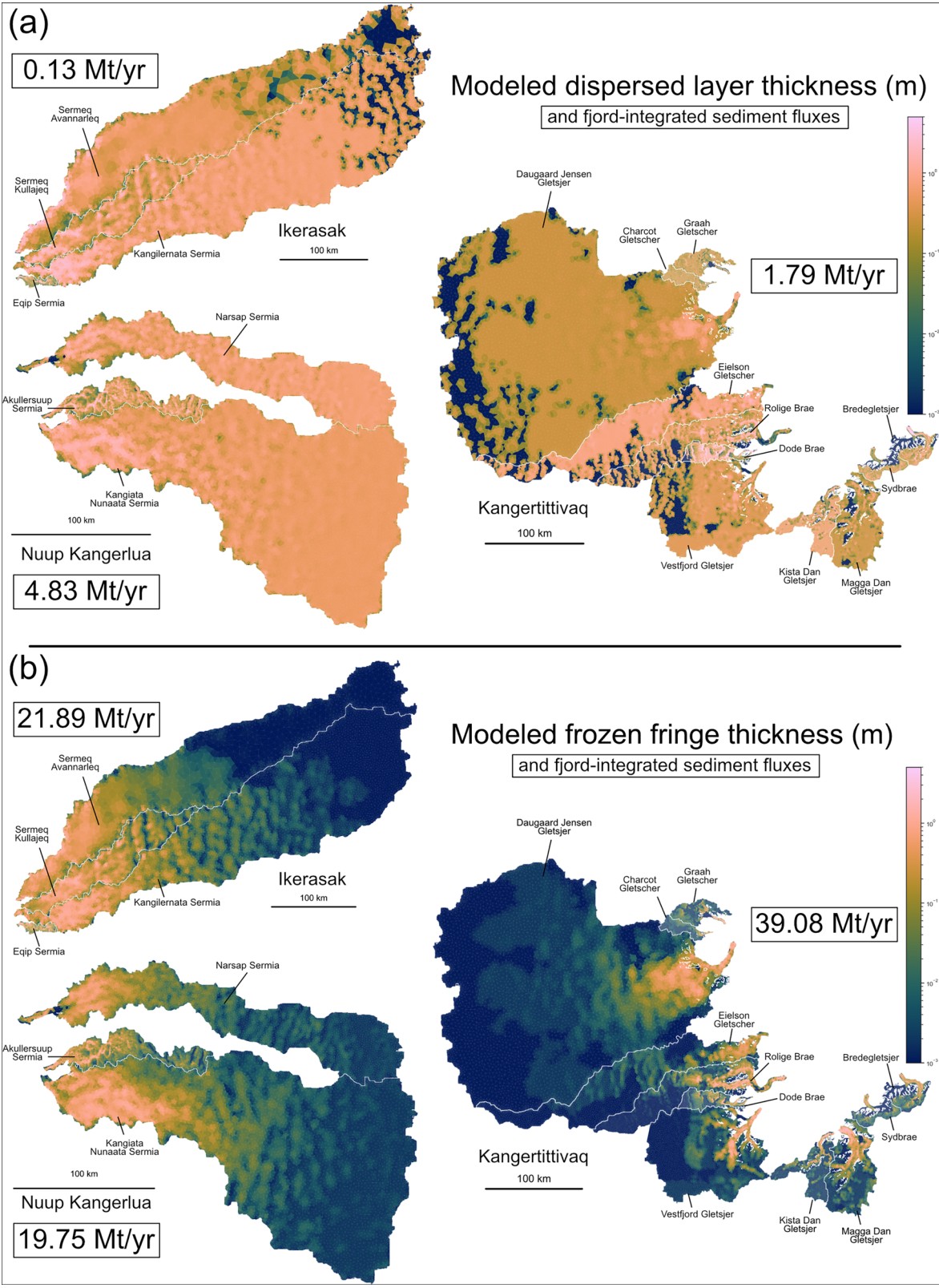

**Fig. 3 | Modeled sediment entrainment.** Modeled catchments in this study. **a** shows dispersed layer thicknesses after 400 years of simulation time. **b** shows steady-state frozen fringe thickness. Annotations indicate the name of each catchment and fjord system. Boxes denote the integrated sediment fluxes in each fjord in Mt a⁻¹ (megatonnes per year).

## Greenland's sediment budget

In log space, we find a statistically significant ($p < 0.001$) relationship between sediment yield and ice yield (Fig. 4). Here, yield is defined as the total flux of ice or sediment divided by the catchment area. Each catchment has a constant ice yield across all model runs, derived from remote sensing observations[31]. The range of sediment yield values is derived from 30 model runs per catchment, varying the most important model parameters across a range of plausible values, for a total of

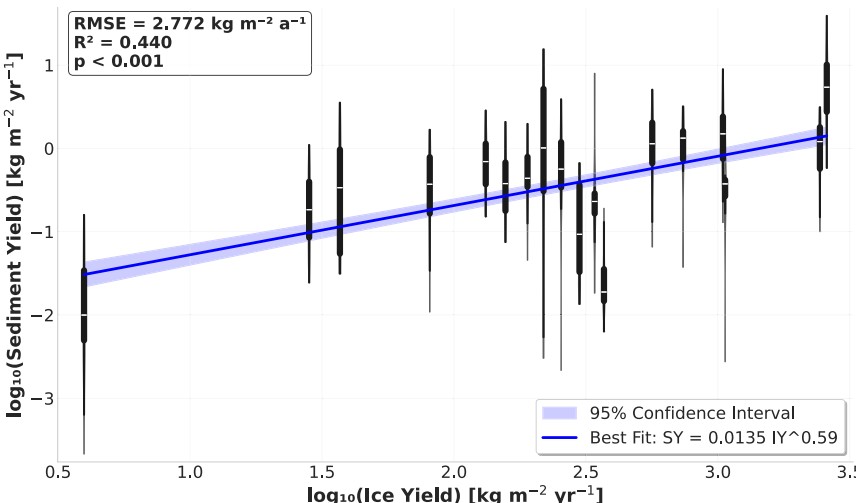

**Fig. 4 | Modeled sediment yields.** The relationship between ice yield (IY) and sediment yield (SY) in log-space, including the best-fit regression plotted with a 95% confidence interval (red line). White lines denote the median sediment yield for each run, with violin plots drawn around the interquartile range, extending to the full range of runs for each catchment (no outliers were trimmed from the plot). Summary statistics include: RMSE (root-mean-square error), $R^2$ (coefficient of determination), and $p$-value (statistical significance).

540 model runs (see Methods). The root mean square error (RMSE) across the dataset is 2.772 kg m$^{-2}$ a$^{-1}$, which corresponds to 1.33 Mt a$^{-1}$ for the median catchment size of the Greenland ice sheet. 80.38% of Greenland's outlet glaciers have an ice yield within the bounds of our modeled catchments, representing 76% of the ice sheet's total solid discharge. From the $R^2$ value, we see that the extrapolated relationship captures 44% of the variability in sediment yield.

Using estimates of ice catchments[32] and ice discharge[31], we predict sediment discharge from basal ice across all of Greenland's outlet glaciers (Fig. 5). We find the greatest sediment fluxes from fast-flowing glaciers with large upstream drainage areas, such as Sermeq Kujalleq, Helheim, Kangerlussuaq, and the Northeast Greenland Ice Stream. The hot-spot regions occur where we see relatively warm, fast-flowing ice and many marine-terminating outlets. Conversely, regional IRD export is weaker in sectors where most outlets have already retreated from the coast (e.g., the southwest) or where most catchments are characterized by lower ice yields (e.g., the north and central-east). From our scaling relationship, we find that the total transport from marine terminating outlets across the entire ice sheet is 454 Mt a$^{-1}$, with a 95% confidence interval from 292 to 716 Mt a$^{-1}$ (that is, +262, −162 Mt a$^{-1}$).

Placing these results in the context of Greenland's source-to-sink sediment budget, we find that IRD delivers approximately one-half to one-third as much sediment as meltwater[3]. We note, however, that some portion of IRD reaches the open ocean, depending on the size of icebergs, the surface water temperature, and the rate at which icebergs topple, preserving debris-rich ice layers from melting out. Because our model does not simulate transport beyond the calving front, these results should be considered an upper bound on deposition in fjords. Additionally, upscaled estimates of fjord accumulation rates have limited access to sediment cores within 10 km of the grounding zone, and so may underestimate the contribution from the lowermost, highly-concentrated basal ice layers[18]. Despite these caveats, sediment fluxes from IRD and meltwater fall within the range of uncertainty of fjord accumulation rates, suggesting that these are the two most significant pathways of sediment transport from the Greenland ice sheet.

## Discussion

Sediment transport from Greenland shapes the morphology of coastlines and grounding zones, plays a key role in biogeochemical cycling in the North Atlantic, and provides a record of paleo-ice sheet activity. We combine observations of IRD with a numerical model of sediment transport, showing that IRD provides a substantial component of Greenland's overall sediment budget. Our findings revise previous estimates of up to 2.88 Gt a$^{-1}$[1-4,19], vastly downward, to 454 Mt a$^{-1}$ (with a 95% confidence interval from 292 to 716 Mt a$^{-1}$). However, our approach is limited by a lack of available data to constrain processes acting beneath ice sheets and glaciers. While we have gathered the most widespread dataset of debris-rich icebergs to date, we have only data from a few fjord systems. The largest sampled catchment is that of the Daugaard-Jensen Gletsjer, but its total sediment flux still does not rival that of Greenland's fastest ice streams. We lack observations from northern Greenland, and have only a few catchments in this study where the majority of the bed is frozen. While our model is well supported by theory and experiments in controlled settings, the subglacial environment is a complex, natural system. Other physical processes that modify pressure, sliding, and sediment availability may exert a strong control on sediment entrainment and IRD export. We have parameterized each component of our model using recent data, selecting parameter values that are most likely to represent the majority of the ice sheet. However, modeling the formation of basal ice layers remains an open problem in glacial physics, and we expect that future work will refine these process models.

Our modeled sediment layers are broadly consistent with field observations of basal sediment entrainment. The model predicts debris-rich basal ice layers on the order of several meters at the termini of most outlet glaciers in the dataset. At land-terminating margins of the Greenland ice sheet, observers have found enriched basal ice layers of 1–3 m[2,33-35], and dispersed basal ice layers extending up to 20 m[35]. We note, however, that dynamic thickening and slow sliding velocities at land-terminating margins might be responsible for such large dispersed layer thicknesses. A compilation of sediment concentrations, also from land-terminating margins of the Greenland ice sheet and several Alaskan glaciers, shows solid ice concentrations of 30–60% and dispersed ice concentrations of 0.1–3.8%[35]. This is consistent with our assumptions about frozen fringe and our measurements of dispersed facies in icebergs.

Analyses of marine sediment cores use a variety of sedimentological properties to identify the presence of IRD, including grain size, angularity, and magnetic susceptibility. Any reconstruction of mass flux will depend on key assumptions about accumulation rates, grain

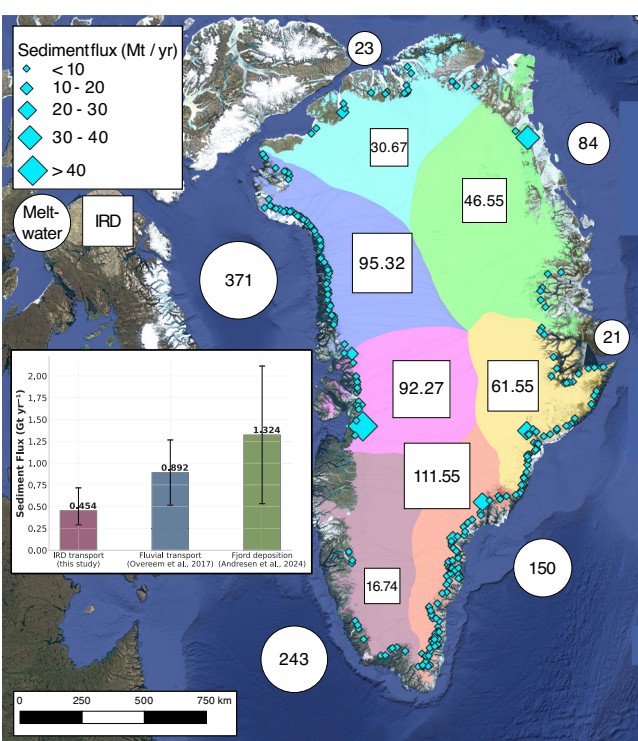

**Fig. 5 | Upscaled ice-rafted debris fluxes.** Upscaled ice-rafted debris (IRD) fluxes for all of Greenland's catchments in Mt/yr (megatonnes per year). Values in boxes are regional IRD fluxes for each of the major sectors of the ice sheet. Values in circles are fluvial sediment fluxes from ref. 3, using the glacial erosion upscaling method. Note that ref. 3 aggregate fluxes by oceanic sink regions, rather than source sectors, and their position has been approximated on this map. The inset barplot shows the bulk components of Greenland's sediment budget: fjord accumulation, fluvial transport, and IRD. Map Data: Google ©2015.

size distributions, and fjord geometry[36]. Our upscaling approach does not distinguish between sediment deposited in the grounding zone and sediment that is transported further into the fjord, and our model of vertical variations in sediment concentration implies that most IRD is deposited early in transit. However, our modeled IRD fluxes are at least within the reasonable range of values that might be expected. In Sermilik fjord, sediment cores taken >30 km from the calving front indicate an IRD flux of at least 1.4 Mt $a^{-1}$ (given a fjord area of approximately $1.4 \times 10^9$ m$^2$)[18]. Similarly, cores from Upernavik Isfjord suggest a fjord-integrated IRD flux of 0.294 Mt $a^{-1}$[37]. In our modeled fjord systems, integrated IRD fluxes from dispersed basal ice are on that order of magnitude, which would be consistent with the hypothesis that frozen fringe melts out closer to the grounding zone. However, this raises important questions about deposition from the frozen fringe. In our model, frozen fringe is required both to form dispersed basal ice layers, but also to insulate them from melt and thus allow export into the fjord. But, sediment cores far into the fjord interior (often >10 km) do not record the elevated fluxes we would expect from the frozen fringe. Future work should focus on collecting observations closer to the glacier margin to improve our understanding of IRD melt-out from the frozen fringe.

Despite these limitations, our model offers important insights into Greenland's source-to-sink sediment transport. Unlike previous approaches, our model resolves vertical variations in sediment entrainment (cf. ref. 3). With the exception of toppled icebergs, basal ice layers melt out from the bottom up once exposed to fjord waters. Vertical gradients in sediment concentration will thus lead to pronounced horizontal gradients in sediment rain-out, with the greatest

deposition rates occurring in the grounding zone. The lowermost layers of frozen fringe carry sediment that is easily identified as IRD, based on coarse grain sizes (> 100 $\mu$m) and subangular to angular particles (see ref. 14). The dispersed layer also contributes to the total amount of IRD exported in a catchment, but is dominated by fine sediment with a much lower proportion of large particles. Along the trajectory of a single iceberg, this dispersed material melts out later than the highly-concentrated frozen fringe. The fine particles are difficult to identify as IRD in marine sediment cores, such that reconstructions of past IRD events may underestimate the dispersal distance, accumulation rate, and timing of sediment fluxes. From the perspective of biogeochemical cycling, it is critical to understand how sediment in the dispersed basal ice layer might differ from the underlying till, since it is far more likely to escape from fjords and reach the open ocean.

Our model of subglacial sediment entrainment also has important implications for long-term projections of tidewater glacier stability. Sediment deposition within the grounding zone inhibits calving, slowing the rate at which glaciers retreat[5,6]. Frozen fringe is uniquely suited to contribute to shoal formation, given its high sediment concentrations and proximity to the ice-bed interface. In our model, sediment fluxes from the frozen fringe are often on the order of megatonnes per year. Conservatively, one Mt $a^{-1}$ of sediment distributed over a grounding zone area of 10 km$^2$ would create specific deposition rates of approximately 5 cm $a^{-1}$, or 0.5 meters per decade. While this is not enough of a topographic barrier to prevent the intrusion of warm Atlantic water[38], previous modeling efforts have suggested that grounding zone sedimentation helps thicken overlying ice, helping to stabilize against stochastic perturbations in local sea level or buttressing[6].

Understanding how glacial sediment transport responds to climate change is an important step towards interpreting paleoclimate archives and forecasting future changes. We identify a significant relationship between a catchment's ice yield (ice flux per unit area) and IRD yield (sediment flux per unit area). Ice yield generally increases with increased mass loss from calving, as the catchment area is much less sensitive to climate forcing. We have observed an increase in ice discharge from Greenland's tidewater glaciers as ocean warming contributes to enhanced rates of iceberg calving[39–41]. Based on our findings, sediment yield should increase alongside ice yield, as long as sediment supply keeps pace with transport processes. This is consistent with paleoclimate studies that show an increase in IRD export during periods of ice sheet retreat (e.g., ref. 42). However, recent work indicates that glacial erosion rates are likely to decrease in the long term as the climate warms, which may complicate this dynamic[43]. On glacial-interglacial time scales, changes in ice sheet configuration could reduce the number of marine terminating outlets, which would force more basal sediment deposits on land. The magnitude and duration of this response remain an open question, with important implications for fjord grounding zone morphology, biogeochemical cycles, and interpretations of IRD in the paleo-record.

## Methods
### Field data
To sample ice-rafted debris in situ, we used small vessels, including inflatable Zodiac boats, in three separate expeditions over 2018–2022. The majority of our samples were taken within 50 km of the terminus, and sometimes as close as 1 km. We also found debris-rich icebergs as far as 120 km away from the terminus in all three systems. To sample, we first identified overturned icebergs that had exposed debris-rich, basal ice layers to the surface. Due to safety concerns, we chose to sample icebergs less than 15 m in freeboard height (i.e., "growlers," "bergy bits," and "small icebergs" in the International Ice Patrol classification). Before taking samples, we removed the outermost 2–3 centimeters of ice. Depending on the time since the iceberg

**Table 1 | Variables, notation, values, and units for model parameters**

| Variable | Notation | Value | Units | Reference |
|---|---|---|---|---|
| Dispersed concentration | $C_d$ | 0.03 | None | Fig. 2 |
| Fringe concentration | $C_f$ | 0.65 | None | Meyer et al., 2023 |
| Till porosity | $\phi$ | 0.35 | None | Meyer et al., 2023 |
| First fringe exponent | $\alpha_f$ | 3.1 | None | Meyer et al., 2023 |
| Second fringe exponent | $\beta_f$ | 0.53 | None | Meyer et al., 2023 |
| Sediment density | $\rho_s$ | 2700 | kg m$^3$ | |
| Ice density | $\rho_i$ | 917 | kg m$^{-3}$ | |
| Ice thermal conductivity | $k_i$ | 2.1 | W m$^{-1}$ K$^{-1}$ | |
| Ice latent heat | $L$ | $3.34 \times 10^5$ | J kg$^{-1}$ | |
| Ice melt temperature | $T_m$ | 273 | K | |
| Fringe entry pressure | $p_f$ | $68 \times 10^3$ | Pa | |
| Characteristic depth | $z^*$ | 100 | m | Pierce et al., 2024 |
| Ice fluidity | $A$ | $2.4 \times 10^{-24}$ | Pa$^{-3}$ s$^{-1}$ | Cuffey and Paterson, 2010 |
| Glen's flow law exponent | $n$ | 3 | | |
| Gravitational acceleration | $g$ | 9.81 | m s$^{-2}$ | |
| Erosion coefficient | $K$ | $2.7 \times 10^{-7}$ | m$^{-1}$ a$^{-1}$ | Herman et al., 2021 |
| Erosion exponent | $l$ | 2 | | Herman et al., 2021 |
| Sheet flow conductivity | $k_w$ | 0.05 | m s$^{-1}$ | Hill et al., 2023 |
| First sheet flow exponent | $\alpha$ | 3 | | Hill et al., 2023 |
| Second sheet flow exponent | $\beta$ | 2 | | Hill et al., 2023 |

overturned, we expect that these layers were enriched from surface melt and are no longer representative of the rest of the debris-rich layer. At each iceberg, we took three samples of approximately 500 mL, at least 1 m apart. We weighed the samples in the field, then transported them in plastic sample bottles back to Boulder, CO, USA, and the University of Copenhagen, Denmark. In the lab, we weighed samples again to ensure no leaking had occurred, and separated sediment by filtering with a 0.7 micron retention diameter and freeze-drying. Samples were freeze-dried on a VirTis BenchTop "K" Series Freeze Dryer to remove all moisture. Finally, we calculated rafted sediment concentrations (RSC) as the ratio of sediment mass to total (ice + sediment) mass.

**Modeled sediment fluxes**

We model sediment fluxes for each individual catchment in our selected fjords. Supplementary Fig. 1 shows a schematic of the sub-glacial sediment transport system, our model components, and source data products. Supplementary Fig. 2 shows boundary conditions and model fields from an example simulation for one glacier (Rolige Brae, East Greenland). We manually delineate the glacier terminus from satellite imagery and the ice basin boundary, and identify the exterior faces of each grid cell at the terminus. The total sediment flux from the catchment is the sum of fluxes through each of these faces. For a single face $i$, the flux is given by:

$$Q_s^i = (H_f C_f + H_d C_d) u_b \ell, \tag{3}$$

where $Q_s^i$ is the sediment flux through that face, $H_f$ and $C_f$ are the height and sediment concentration of the frozen fringe, $H_d$ and $C_d$ are the height and sediment concentration of the dispersed layer, $u_b$ is the glacier's sliding velocity, and $\ell$ is the length of the face. The total $Q_s$ at a catchment is the sum over all $Q_s^i$ at grid faces across the terminus, which we delineate by hand. Given our sparse observations of frozen fringe, we follow previous work and assume $C_f$ is related to an esti-mated average till porosity beneath the ice sheet. For the dispersed layer, we use the mean RSC value from our field observations. Sliding velocity is discussed below. The remaining terms, $H_f$ and $H_d$, are the primary outputs from our sediment transport model.

We model the transient evolution of frozen fringe and dispersed basal in each catchment. The governing equations for each layer are:

$$\frac{\partial H_f}{\partial t} + \nabla \cdot (u_b H_f) = \frac{-m - V}{\phi S}; \tag{4}$$

$$\frac{\partial H_d}{\partial t} + \nabla \cdot (u_b H_d) = \frac{3k_i}{\rho_i L} G_i. \tag{5}$$

The source term for the frozen fringe depends on the specific basal melt rate $m$, the rate of vertical heave $V$, the till porosity $\phi$, and the saturation of the fringe $S$[25]. The dispersed layer growth rate depends on the thermal conductivity of ice $k_i$, ice density $\rho_i$, the latent heat of fusion for ice $L$, and the thermal gradient $G_i$[27]. Where frozen fringe is absent, we allow the dispersed layer to melt, adding $-m$ to the right-hand side of eq. (5). We use the frozen fringe model presented in ref. 25, with the parameters shown in Table 1, except for key controls varied as part of our uncertainty quantification (Supplementary Table 1). The heave rate, which governs vertical sediment motion into and out of the fringe, is a nonlinear function of the effective pressure $N$, non-dimensional undercooling $\theta$, and the till porosity $\phi$. Generally, large effective pressures lead to negative heave rates, and thus frozen fringe growth. The heave rate is given by Eq. (13) of ref. 25, with $\alpha_f = 3.1$ and $\beta_f = 0.53$ informed by ref. 24, and all other parameters for the heave model as shown in ref. 25.

The dispersed layer growth rate depends on the thermal regime at the base of the ice sheet. We assume that the bed is frozen in regions where basal melt is absent, and ignore any dispersed layer growth in these regions. In regions where basal melt is positive, we assume the bed is thawed, and there is a layer of temperate ice, even if quite thin, at the base of the ice sheet. The temperature at the top of the frozen fringe $T_l$ is:

$$T_l = T_m - \theta(T_m - T_f), \tag{6}$$

where $T_m$ is the melting point of ice, $\theta$ is non-dimensional undercooling, and $T_f$ is the temperature at the bottom of the fringe. This base temperature is set by a threshold pressure for ice to infiltrate sediment, denoted $p_f$, such that $T_f = T_m - \frac{p_f T_m}{\rho_i L}$. Here, $p_f = 68$ kPa, based

on a characteristic grain size of 0.15 mm (medium sand). Then, the temperature gradient $G_i$ above the frozen fringe is given by:

$$G_i = \frac{T_l - T_m}{z^*}, \tag{7}$$

where $z^*$ is the characteristic depth of the temperate ice layer (Pierce et al., 2024). In reality, there is likely spatial variability in $z^*$, but here we vary the parameter from 10-200 m, given our lack of additional constraints. While conservative, this range of values yields debris-rich ice layers on the order of meters, matching our field observations, and thus represents a lower bound on dispersed layer growth rates.

## Sliding velocity

To model the long-term mean sliding velocity, we assume that basal slip and internal deformation are the two primary components of velocity across the majority of the ice sheet. Then, the residual of observed surface velocity, $u_s$, and deformation, $u_d$, provides a first-order approximation of sliding velocity, $u_b$[30]:

$$u_b = -(\nabla s) * max[|u_s - u_d|, 0], \tag{8}$$

where $\nabla s$ is the ice surface gradient. Where modeled deformation velocity is greater than surface velocity, we simply set the sliding velocity to zero. We set the direction of sliding velocity opposite to that of the surface gradient, such that ice flows downslope. Over large spatial scales, deformation velocity is described by the shallow ice approximation[30]:

$$u_d = \frac{2A}{n+2}(\rho_i g \nabla s)^n H^{n+1}, \tag{9}$$

where $A$ is a coefficient describing ice fluidity, $n$ is the exponent in Glen's flow law, and $H$ is the ice thickness. We take the standard assumptions that $n = 3$ and let $A$ vary from $3.5 \times 10^{-25}$ to $2.4 \times 10^{-24}$ Pa$^{-n}$ s$^{-1}$, corresponding to ice between 0 and $-10°$ C[30]. This range for $A$ is conservative, in that it may overestimate ice fluidity in colder regions of the ice sheet, leading to lower sliding velocities and less geomorphic work. To help uphold the assumption of a shallow aspect ratio, we smooth the surface elevation using a Gaussian kernel with a width roughly equal to 3 times the ice thickness.

Our model does not allow sediment entrainment in the frozen fringe unless there is an available supply of material in the underlying till layer. At each time step, we update the thickness of the till layer via erosion, then remove material as it is added to the frozen fringe. We do not consider lateral motion in the till layer. We model erosion using a power law relationship from an empirical fit to a global compilation of glacial erosion rates[43]. The erosion relates to glacial sliding as:

$$E = K|u_b|^l, \tag{10}$$

where $E$ is the erosion rate, $K$ is the rate coefficient, $|u_b|$ is the magnitude of sliding velocity, and $l$ is a tunable exponent. Following the results from ref. 43, we co-vary $K$ and $l$ along a best-fit relationship identified for Greenland (see Supplementary Table 1). Eroded sediment is added to the till layer, where it becomes available for entrainment in the frozen fringe. While process-based models of erosion may perform better in catchments with more constraints on the subglacial geology, these formulations often rely on a greater number of unconstrained parameters, which are difficult to apply uniformly across many different catchments.

## Subglacial effective pressure

Another important boundary condition is the effective pressure, defined as the residual between ice overburden pressure and water pressure beneath the glacier. We often model the subglacial drainage system as a combination of channelized (efficient) and distributed (inefficient) components[44–47]. These models are computationally complex, rely on careful parameterizations of both systems, and are unlikely to reach a numerically stable configuration for every catchment[44]. For the purposes of this model, we only solve for pressure in the distributed drainage system. This simplification follows from two key assumptions. First, the channelized drainage system occupies much less subglacial area than the distributed system. Frozen fringe only forms where the base of the glacier is in contact with sediment, not over active conduits. Second, while channels do modify the shape of the pressure field, they do not modify the large-scale effective pressure far afield from the channel itself[45]. Over decades to centuries, changes in channel position and flow conditions will average out any local changes to the pressure field that we are unable to resolve with this model. Importantly, channelized flow is incorporated into the basal meltwater data used here[48], ensuring that meltwater still contributes to heating basal ice and increasing sediment deposition.

To model the distributed drainage system, we iteratively solve for the thickness of sheet flow and hydraulic head until the system achieves a steady-state. Sheet flow evolves by:

$$\frac{\partial h_w}{\partial t} - \nabla \cdot q = m, \tag{11}$$

where $h_w$ is the height of sheet flow in the distributed system, $q$ is the discharge, and $m$ is the basal melt rate. Then, we use Darcy's law to formulate an elliptic partial differential equation for discharge:

$$q = -k_w h_w^\alpha |\nabla \psi|^{\beta-2} \nabla \psi, \tag{12}$$

where $k_w$ is the hydraulic conductivity, $\alpha$ and $\beta$ are tunable exponents, and $\psi$ is the hydraulic potential. Basal water pressure $P_w$ is related to hydraulic potential by $P_w = \psi - \rho_w g b$ for water density $\rho_w$ and bedrock elevation $b$. We follow the results from ref. 49 and take $\alpha = 3$ and $\beta = 2$, corresponding to laminar flow, which provides a better approximation of the long-term, spatially-averaged water pressure than turbulent parameterizations. Along the boundaries, we set $\psi = 0$ at outflow nodes and $\nabla \psi = 0$ around the rest of the catchment. We solve the system of equations iteratively using a backward-time, finite volume scheme, until hydraulic head converges to a steady state.

## Numerical implementation

We rely on community data products to provide boundary conditions for our sediment transport model. Ice catchments were delineated by ref. 32, using velocity mosaics in fast-moving regions (> 100 m a$^{-1}$) and surface slope in slow-moving regions (< 100 m a$^{-1}$). We use ice flux estimates from ref. 31, which are based on time-varying ice thickness and velocity maps and automated identification of termini and flux gates. For our flux scaling, we calculated background ice discharge as the average discharge from 2018-2023. We obtain ice sheet thickness and inferred bedrock geometry from BedMachine v5[50], which uses a mass conservation approach. Basal melt rates were estimated by ref. 48, based on an energy balance and a synthesis study on Greenland's likely basal thermal state[51]. Finally, we use the MEaSUREs ITS_LIVE 120 m surface velocity mosaic, derived from remote sensing measurements from 2023[29].

For each catchment, we generate an unstructured mesh with the Triangle software package[52] and Landlab gridding tools[53]. The mesh has an average of 10,000 elements for each catchment, with the grid resolution increasing towards the margins of the domain. The nominal resolution for the largest catchments is between 1 and 5 km$^2$ per grid cell. We solve the governing equations using a finite volume discretization in space, and a combination of forward and backward time-stepping methods depending on the numerical behavior of each

equation. On a standard research laptop, the model takes several hours to run a single catchment. Due to computational limits, a simulation of the full ice sheet would not be feasible without considerable attention to optimization.

To quantify the uncertainty in our results due to unknown or under-constrained model parameters, we conduct a limited Monte Carlo analysis for each catchment. From prior sensitivity analysis[54] and preliminary investigation of our model results, we identified the 9 parameters with the greatest impact on modeled IRD fluxes. We then conducted 30 simulations for each catchment, selecting these variables at random from a Gaussian distribution fit to a range of physically plausible values (Supplementary Table 1). Dispersed sediment concentrations were sampled from a Pareto distribution fit to the field data. We did not remove any model runs from the final dataset, such that the sediment yield results used to construct our upscaling relationship and all summary statistics are based on the full array of 540 simulations. Finally, we constructed a 95% confidence interval around the best-fit relationship using a two-sided t-test with the standard error of the fit given by the `linregress` function in the `scipy.stats` package.

## Data availability

Data from IRD samples are available at https://doi.org/10.18739/A2NC5SF41[55]. Ice catchment delineations are available at https://doi.org/10.7280/D1WT11. Ice discharge data is available at https://doi.org/10.22008/promice/data/ice_discharge/d/v02. Ice and bedrock geometry from BedMachine is available at https://doi.org/10.5067/GMEVBWFLWA7X. MEaSUREs ITS_LIVE surface velocity is available at https://doi.org/10.5067/6II6VW8LLWJ7. Basal melt rate maps are available at https://doi.org/10.22008/FK2/PLNUEO.

## Code availability

The software used to produce the results in this paper is open-source and archived at https://doi.org/10.5281/zenodo.14171812.

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

## Acknowledgements

E.P. and I.O. were supported by funding from National Science Foundation awards for OpenEarthScape (2104102) and the Community Surface Dynamics Modeling System (2148475) and a University of Colorado Boulder Research and Innovation Seed Grant. B.H. was supported by the Danish Centre for Marine Research.

## Author contributions

All authors contributed to the conceptualization of the project, interpretation of results, field sampling, and data analysis, and review/editing of the text. E.P. was responsible for model development, visualization, and writing the original draft. Both I.O. and B.H. were responsible for supervision, field methodology, data curation, and funding.

## Competing interests

The authors declare no competing interests.
