## [Transparent Peer Review file · Nature Communications]

Sediment transport by Greenland's icebergs

Corresponding Author: Dr Ethan Pierce

Version 0:

Reviewer comments:

Reviewer #1

(Remarks to the Author)

To quantify iceberg rafted debris in Greenland's total sediment budget, Pierce et al combine field data from 210 iceberg samples across three fjords with a numerical model of erosion and entrainment to estimate IRD fluxes. They find that sediment concentrations span a wide range (0.1–45% by mass), and discuss this in context of frozen fringe sediment and dispersed basal sediment. They find a first-order relationship between ice and sediment yield and estimate that icebergs export ~416 Mt/a of sediment, or roughly one-third of Greenland's total sediment flux.

Interestingly, the estimated iceberg sediment flux presented in this manuscript is significantly higher than previous estimates reported by Overeem et al., 2017 and exceeds the values anticipated in the discussions in Andresen et al., 2024. However, I have concerns regarding both the field observations used to constrain the model and the approach taken to validate this elevated flux estimate.

Field observations: Hasholt et al, 2022 describes a method to collect samples of ice from icebergs systematically. They point to their 97% uncertainty range based on only 12 icebergs – and their method is meant for future sampling to provide more realistic values as data collection hopefully increases. They write: We present an estimate of the transport related to calving of ~100 million tons per year with a range from 0.3 to 200 million tons. We are fully aware that 12 samples of icebergs are not sufficient to constrain the transport of sediment reasonably accurate, but we have performed the calculation to illustrate the effects of biased sampling. The simultaneous meltwater-driven suspended sediment fluvial contribution is 21–30 million tons (from table 1 in Overeem et al. 2017). The latter number refers to the meltwater sediment flux from the entire Scoresbysund region. For comparison Overeem et al. 2017 find that iceberg sediment accounts to c. 1% of that contributed from melt water (their table 1) for entire Greenland. While I acknowledge the considerable effort involved in collecting and analyzing data from 210 icebergs, I am not convinced that increasing the sample size alone provides greater confidence in the upscaled estimate. Given the inherent variability in iceberg sediment content and the challenges in achieving unbiased sampling, I believe the current dataset is insufficient to support a robust extrapolation to the ice sheet scale. This is not to discount the value of the fieldwork, which is clearly substantial, but rather to caution that the upscaling appears premature. As a first step, a more informative approach would have included a detailed characterization of the sediment contained within the sampled icebergs—particularly grain size distributions and lithologies—as well as descriptions of sediment banding features, including band thickness, contact relationships, and three-dimensional morphology. This is only partly touched upon – but a systematic report and analysis of the iceberg data from the 210 bergs, including photographs, would be very valuable for our understanding of Greenland's 'muddy bergs'. Greenland's iceberg population is indeed highly heterogeneous, and a deeper understanding of this variability is needed before attempting large-scale quantification of IRD fluxes.

Validation: The only validation provided in the manuscript is a comparison with sediment flux estimates from Overeem et al., 2017 and Andresen et al., 2024. Pierce et al. argue that the difference between the two studies reveals an unaccounted sediment fraction attributable to icebergs, noting that Overeem reports suspended sediment flux, whereas Andresen reports total sediment flux. However, this interpretation is not correct. Andresen et al. estimate total sediment flux only from marine-terminating glaciers at 1.324 ± 0.791 Gt/yr, while Overeem et al. report two estimates of suspended sediment flux from both marine- and land-terminating glaciers: 0.892 Gt/yr (erosion-based) and 1.28 Gt/yr (meltwater discharge method) (and both studies btw point to a low contribution from iceberg-derived sediment). So, the difference between the two studies clearly cannot be used as a proxy for iceberg sediment flux. A short summary of the two papers, for the readers not deeply acquainted with these studies, would have clarified this.

Moreover, the methodologies in these studies differ significantly. Andresen et al. rely on sediment core accumulation rates, while Overeem et al. use remote sensing and modeling of meltwater fluxes. Both approaches involve large uncertainties, which are acknowledged in their respective discussions in these papers. However, the broad agreement in their flux estimates, despite such different methodologies, suggest they are both on the right track. But their differences reflect these methodological differences and limitations—not a missing ~430 Mt/yr of sediment. For example, Overeem's method is limited by sparse data near deep glacier termini, which alone could account for substantial underestimation. Likewise, Andresen study is limited by not having sediment cores right next to the glacier margins.

The manuscript's interpretation of this discrepancy as iceberg-derived sediment is therefore problematic, especially as it forms the sole basis for validating the model's sediment yield.

A question that comes to mind here is: Why have they not compared with sediment core data from Greenland's fjords and shelf presenting iceberg rafted debris fluxes?

Some minor comments

Line 33 and 34: Some of these studies also point to plume sediment as a shoal-builder, in particular the studies of temperate Alaskan glaciers.

Line 36: heightened levels of chlorophyll are linked with sediment in the Southern Ocean where iron is a limited nutrient. Around Greenlandic icebergs local upwelling of macronutrients from Atlantic-sourced water may be the cause of heightened chlorophyll, and not the elements contained in the sediment. So, it would be correct to mention these observations goes for the Southern Ocean.

Line 113: what are the characteristic sediment grain size distribution in the frozen fringe sediment, and in the dispersed basal ice facies according to the referenced literature? And are these based on in situ observations by landterminating margins? This information is highly relevant when linking the sediment sampled from icebergs with the sediment entrained underneath the glacier. Do the grain size distributions match?

Line 154. What is the role of subglacial meltwater derived from surface melt? Does channelization of meltwater near the glacier margin influence the sediment content at the ice-bed interface? This likely represents an important process through which basal sediment is transferred into the meltwater plume. If this mechanism is not accounted for, estimates of sediment yield from icebergs may be significantly overestimated. While these processes are reported on in the methodology section is not mentioned in the discussion and it is not clear to me how IRD and plume sediment is partitioned in this study.

Line 156-157: How do the erosion rates compare to previous reported estimates of erosion rates from Greenland catchments? I believe the ones reported by Pierce et al. are 100-fold higher than previous estimates?

Line 165: is this not expected when the same surface velocity and terminus width are primary variables in calculations of both sediment yield and ice yield (data in Mankoff)? The derivation of ice yield is reporting in the methodology section, but there is no reference to this on the Figure 4C.

Line 245-248: I suggest to delete this statement. Sea level will decrease by Greenland's glacier due to isostatic rebound and reduced gravitational pull; it will not increase here. The question is if shoaling can prevent incoming warm Atlantic waters and 50 cm of sediment per decade would do nothing here.

Line 228: grain sizes and angularity....would be great to be more specific here.

Line 230: how is that known?

Line 232-234: to be clear to a broader audience – specify what is meant by reconstructions. But be also aware that reconstructions of iceberg production and dispersal changes in the past are based on the coarse fraction (for the reasons mentioned by Pierce et al.).

Line 235-237: I agree, and for this reason grain size distributions of the samples would have been informative.

Line 250-252: it says: " This assumption holds if the system is limited by supply- that is, if the glacier has the capacity to transport all available sediment." Should it not be "is not limited by supply"?

(Remarks on code availability)

Reviewer #2

(Remarks to the Author)

Review of "Sediment transport by Greenland's icebergs" by Pierce et al.

In this paper, the authors quantify the volume of sediment exported by icebergs calved from Greenland's glaciers. They focus on three glacial systems and extrapolate their findings to the entire ice sheet. Their interpretation is based on numerical modeling that encapsulates current understanding of sediment storage in glaciers and associated erosion/transport processes. The main finding is that approximately one-third of sediment export from Greenland occurs via icebergs. This represents an important contribution, and I hope to see it published.

That said, I have a few recommendations which I hope will help the authors clarify certain points:

- 1) While I understand the different components of the numerical model, it took me some time to fully grasp its structure. A schematic illustrating the model's elements would be very helpful. This is important because the model underpins much of the interpretation—it includes calculations of sediment flux, transport by frozen fringes, and dispersion, which are all clearly presented. However, the mechanism by which erosion feeds sediment into the ice layers remains unclear to me, even though equations 4 and 5 incorporate erosion as a source term. As a result, the discussion on supply-limited vs. transport-limited regimes is somewhat unclear.
- 2) The authors use a single set of values for the erosion parameters. While I agree with the decision to keep the erosion model simple, I would be interested to see how sensitive the results are to the choice of K_g and I . Does this affect the results shown in Figure 4a—for example, the ratio of erosion to transport? Additionally, how much sediment could accumulate if erosion continues to outpace transport over centuries, millennia, or even tens of thousands of years?
- 3) The authors show that sediment content in icebergs follows a long-tail distribution. If I understand correctly, this is based on the assumption that the icebergs are representative of the basal layer. But are we certain that icebergs aren't just fragments of the ice front? In other words, could they be breaking through the middle of the sediment layer? I would expect such a process to also produce a long-tail distribution. How can we be confident this is not the case?
- 4) The authors present sediment export in terms of volume. One of the most striking results is the correlation shown in Figure 4c between ice yield and sediment yield. I wonder what the relationship would look like if plotting ice velocity vs. sediment yield instead?

Specific comments:

Abstract, line 28 – The paper doesn't address how sediment transport may evolve (or has evolved) in response to climate change. That said, I would be very interested to read any projections or hypotheses the authors might offer. Will we see more sediment delivery or greater sediment storage within glaciers?

Line 42 – It would be helpful to also indicate the total sediment volume here.

Line 73 – The error bar is missing after "416 Mt."

Lines 90–93 – How can we be certain the sampling captures the entire basal layer rather than just part of it?

Line 157 – These erosion rates are quite high. How sensitive are they to the model parameters?

Figure 3 – What causes the patchy appearance in the data?

Line 213 – Is this really a controlled setting?

Lines 259–260 – I suppose this depends on the model used?

Line 273 – What is the expected behavior of erosion over time?

Line 363 – It's unclear how erosion modifies the till layer. Some clarification would help.

(Remarks on code availability)

Reviewer #3

(Remarks to the Author)

Pierce et al. present modelled estimates of ice-rafted debris (IRD) in three fjord systems. Model outputs from 11 glacier basins are used to derive an empirical relationship between iceberg sediment yield and an existing estimate of grounding line discharge. This relationship is then used to estimate the total sediment yield from Greenland's solid ice export. This result represents the 'missing piece' in Greenland's total sediment production, so the result is important. Reflective of this importance, I anticipate that the community will readily adopt the values for IRD presented in the manuscript. In anticipation of this, it is essential that the assumptions used to derive those values are clearly presented, the sensitivity of the modelling to various parameter choices are comprehensively quantified and model validation is performed wherever possible. In its present form, the manuscript could (and should!) do all of these better, which leaves the reader lots of reasons to be sceptical of the presented IRD values. My major comments below describe where I think the manuscript falls short in these respects, and provide some suggestions of how to make the result as robust, transparent and convincing as possible.

Major comments (in no particular order)

- 1) There are many parameters in your model for which the appropriate value is unknown or poorly constrained, yet no sensitivity tests are presented to examine the influence of parameter value choices on modelled sediment thickness. Related to this, one value for dispersed sediment concentration is used in the simulations, yet visual examination of Figure 2 suggests significant differences between fjords – please quantify this difference and justify the use of a fixed value. The manuscript even states on lines 95-97 that sediment concentrations vary significantly among nearby samples and that this variability needs to be accounted for in models, yet a fixed value is still used and the impact of this choice on the results is not presented. Please provide a thorough sensitivity analysis of all uncertain parameter values and quantify how they affect the modelled IRD in each of the fjords with field data, as well as the reconstructed IRD for all of Greenland.

2) As far as I can tell, your simulations and scaling up assume that there is porous till everywhere in all of Greenland's tidewater glacier basins. This assumption is not clearly stated and the impact of the assumption is not quantified. There are very few constraints on the characteristics of the substrate in tidewater glacier basins, but boreholes in land-terminating catchments indicate extensive areas of bedrock (<https://agupubs.onlinelibrary.wiley.com/doi/full/10.1002/2017JF004201>). Given this uncertainty, it seems unreasonable to assume the extreme end-member that there is till everywhere. This is clearly an important assumption because the frozen fringe appears to provide the vast majority of sediment transport in your model (Fig. 4). Please either provide convincing justification for this choice or modify your assumptions in the simulations based on the little available evidence in the literature. Whichever you choose, please state your assumption clearly and, if reasonably practicable, quantify the impact of that choice on your results.

3) As presented, the only model validation is the Greenland-wide comparison with the residual IRD in Overeem et al. (2017), which in turn relies on scaled-up modelled ice sheet runoff (which is known to substantially misdiagnose measured runoff) from both the GrIS and surrounding ice caps, and, if I understand it correctly, also does not account for IRD that escapes from fjords in icebergs. The presented validation does not remove the effect of sediment export from surrounding ice caps in the Overeem et al. (2017) estimate, which would clearly affect the residual IRD value. I appreciate that there is very little data available for validation, but could you also use the Overeem et al. (2017) estimate on a catchment-by-catchment basis and compare those with your simulated sediment export? Could you directly compare your simulated sediment layer thickness at the terminus with those observed by you and others in the field? (Related to this, it would be helpful if your modelled sediment layer thicknesses were presented clearly – see comment below). If the sediment concentration evolves within the model, could those values be compared to your measured values? It's not clear that the model is internally consistent: do the modelled erosion and transport rates combine to provide the observed sediment concentration values? In addition, could you simulate an additional glacier basin (or ideally several basins) and compare the resulting modelled IRD to the reconstructed IRD based on grounding line discharge? I appreciate that considerable effort is required to create new model domains and a small amount of computational time (a few hours) is required to run the model, but this would make the upscaling more convincing.

4) The reconstruction of Greenland-wide IRD could be presented more transparently. Firstly, please provide details regarding the estimation of the uncertainty bounds on the fit – this uncertainty should be updated to reflect the sensitivity analysis recommended above. Secondly, please provide the RMSE (or a similar metric) for the performance of the fit against the modelled basin IRD fluxes. Third, please describe the distribution of ice yield amongst the unmodelled catchment within the bounds of the fit – i.e. for how many basins do you need to extrapolate the fit? What fraction of the total IRD do they provide? Over what time period is grounding line discharge averaged to produce the fit and reconstruction and how does that differ from the average discharge during the period over which the ITS_LIVE 120 x 120 m ice surface velocity was generated? By producing the fit as presented, you are implicitly assuming that your simulations capture all of the variability of sediment layer thickness around Greenland and that all of the variability in total sediment yield is due to spatial variations in sliding speed. You should back this up with your model results – you have modelled sliding speed and modelled sediment layer thickness, so use those to demonstrate that sediment yield mostly varies with sliding speed not sediment layer thickness and show that sediment layer thickness at the terminus does not have a consequential range. You have presented an estimate of Greenland-wide IRD, and in many ways this is the value everyone wants to see, so either make it convincing or make it clear that it would not be prudent to attempt it because the uncertainties are so large.

5) Data availability: the model output and reconstructed IRD estimates for each glacier basin, glacier region, and summed across the ice sheet (as shown in Figure 5), should also be made available. These are the primary outputs from this manuscript.

Minor comments

Line 25: Please clarify what the values in parenthesis represent. Or it may be clearer to just provide the range initially, then the most likely value.

Line 31: "Originating far inland" is, I think, inaccurate. Does IRD not accumulate up to the terminus?

Line 35: This is true for Antarctic icebergs – indeed, each of the references given here are of studies focusing on the Southern Ocean. The subject here is Greenland, where iron is less frequently a limiting nutrient and any increase in biological activity around icebergs may be due to upwelling of nutrient-rich Atlantic Water into the photic zone. You should either clarify that this sentence refers to Antarctic icebergs or modify the sentence to reflect the Greenland/North Atlantic literature.

Lines 41-43: these two values conflate the results of two studies. I suggest you modify the first sentence to present the total sediment fluxes from both Andresen et al. (2024) and Overeem et al. (2017), and make it clear that the runoff-only value is based only on the results of Overeem et al. (2017).

Line 141: "cm a-1" isn't an accurate representation of speeds in the majority of the ice sheet interior. The ITS_LIVE 120 x 120 m mosaic suggests most areas in the interior have speeds >10 m a-1 and GNSS measurements 140 km from the margin showed speeds of ~50 m a-1.

Line 152/153: Can you back up this statement about the frozen fringe with numbers? This is a key assumption in section 4, so I think it is important that it is quantified.

Line 156: The use of "maximum" here is confusing. It would be clearer to just say that the highest catchment-average erosion rates are X and Y at Eqip Sermia... etc. Those two erosion rates also seem unrealistically fast.

Lines 171-173: The meaning of this sentence is a little hard to figure out, or how you determined that the model is likely underestimating the true IRD export. What do you mean by "choice of exactly where to measure sediment fluxes"? Since your model is in steady-state, do you not just integrate the transport rate? If you have to specify some model grid cells near the terminus, then perhaps you can be more specific than just "highly sensitive" e.g. by what proportion does it change if you

sample one pixel upstream?

Line 272/273: I don't think your results justify this statement. The statement implies that (1) outlet glaciers will continue to accelerate in future; (2) as outlet glaciers accelerate, erosion rates will increase; (3) that your relationship between ice yield and sediment yield is robust at a single accelerating glacier over time, rather than only across multiple glaciers which are assumed to be in steady-state, and that the increased sediment production is greater than the uncertainty in the fit; (4) even though the system is transport limited, increased erosion will also lead to more transport.

Figure 4c: I recommend changing the units on this figure to Gt or Mt per year, rather than providing them per unit area. The metric of interest here is the export of sediment across the calving front, so it does not make sense to provide it per unit area.

Figure 4c: Are there points missing from the Kangerittivaq region on this graph? I am assuming that each point corresponds to one of the labelled outlets in Figure 3 – there are 11 labelled outlets but only 8 points on the graph.

Reviewed by: Benjamin Davison

(Remarks on code availability)

I have not reviewed the code. From a quick look at the zenodo repository, it looks like the results would be possible to reproduce with considerable effort. I could not see the model output in the repository, which seems like a major omission. As per my major comments above, please provide the model fields from the target basins, the IRD export from those basins as provided by the full model, and provide the reconstructed IRD for all basins, all ice sheet regions, and the ice sheet as a whole.

Version 1:

Reviewer comments:

Reviewer #1

(Remarks to the Author)

(Remarks on code availability)

Reviewer #2

(Remarks to the Author)

I sincerely apologize for the delay in this second round of review.

After carefully reviewing the manuscript and the response letter, I can confirm that all my concerns, particularly regarding the description and use of the model, have been addressed.

I look forward to seeing this work published.

(Remarks on code availability)

I did not review the code.

Reviewer #3

(Remarks to the Author)

This is my second time reviewing this manuscript. In my first review, I recommended that Pierce et al. provide a clearer description of their assumptions, analyse the sensitivity of their results to model parameter value choices and, wherever possible, validate their results. Their response is one of the most thorough, informative and collegiate responses to a set of reviewer comments that I have seen. I was grateful especially for the additional explanations in response to my comments where I had misunderstood the work (controls on sediment layer thickness and Fig. 4), for which I apologise – it seems you have been similarly collegiate in your response to another reviewer's interest in grain size and lithology.

I think the revised manuscript represents a significant improvement on existing estimates of IRD from Greenland's tidewater glaciers, which will be of great interest to the community and will help guide future research. I have a few minor suggestions that I think would help to further improve the manuscript. Once those points are addressed to the editor's satisfaction, I wholeheartedly recommend the manuscript for publication.

Minor comments

Title: consider "Sediment transport by Greenland's tidewater glaciers" or similar. Transport by icebergs makes me think of iceberg drift after calving.

Line 47: "Gt of sediment per year" – consider explicitly stating that this is sediment from IRD and meltwater, if you have spare words.

Line 57: Sorry, but I don't quite follow how the preceding introduction of Andresen et al. and Overeem et al. points to large

uncertainties in the contribution of sediment transport by IRD. As mentioned in the revised text, the given numbers do have large enough uncertainties that they could be reconciled, so I wonder if it would help readers if you were to open the door to IRD as a potentially significant contributor earlier in the paragraph. Would be it correct to add a caveat on line 50 to introduce the idea that, despite the relationship between runoff and core accumulation rates, IRD must still be a significant source? Or provide some nuance about the spatial distribution of the cores vs expected IRD locations?

Lines 107:110: in my previous review of this manuscript, I questioned the use of a fixed value for sediment concentration in the model given this statement about highly variable sediment concentrations being important for modelling sediment flux. I appreciated your response explaining the cause of this variation between samples and I wonder if some of that reasoning should be transferred to this section of the manuscript, so as not to cause similar confusion or scepticism for readers. Essentially, I think the choice of a fixed sediment concentration is a simplification of reality that is required to make the problem computationally tractable, but, as written, you are making this choice seem scientifically unjustifiable.

Line 150: Can you describe how you defined steady-state? E.g. monthly fluctuations of less than x?

Line 153: I think it would be better if you named the glaciers here, rather than using "southwest" etc, otherwise you imply that they are representative of all of southwest Greenland etc.

Line 154-156: re time to reach steady-state - I think this would be better placed at the end of the previous paragraph.

190-192: I wonder if it would be clearer to remove this text. Given the results and conclusions focus on marine terminating glaciers, it's a little confusing to discuss what are now land-terminating outlets. It's also not clear what you mean by a "less efficient" catchment.

Line 214-215: consider providing again your estimate.

Line 231: this appears to contradict the statement on line 214/215 because it's not immediately clear that you're no longer discussing Greenland's overall IRD flux. I think it would be helpful if you specified which aspects of the model results you are referring to in the opening sentence.

Line 297-299: I think it would be more accurate to say that "we have observed an increase in mass loss from Greenland's tidewater glaciers..." because discharge hasn't increased at the ice sheet-scale since 2005. If you're tight for words, you could (and maybe should) not even mention mass loss and just go straight to ice discharge, which would let you remove the clause after the comma... Then also modify the following sentence about sediment yield increases accordingly.

Line 403 and elsewhere: you state single parameter values but you tested a range of values. I think you should provide the range here as well as in Extended Data Table 2.

Figure 3: I think the panel labels (a,b) may be missing, or I can't see them. I think the top panel showing dispersed layer thickness should use the same colormap as the bottom panel – it looks like they are on the same scale after all and it's odd to use a diverging colormap for non-diverging data. As with Figure 5, it would be great to include the uncertainties as well as the central value.

Figure 5: this is a really effective figure! My only suggestions are to (1) move the bold text in the barplot from the top of the error bars to the top of the shaded bar; (2) add lines delineating the ocean sink regions used, and; (3) consider adding uncertainties to the boxes for the ice sheet sectors and ocean regions.

Reviewed by: Benjamin Davison

(Remarks on code availability)

Thank you for giving us the opportunity to submit a revised draft of the manuscript “Sediment transport by Greenland’s icebergs” for publication in *Nature Communications*. We greatly appreciate the time and effort spent by you and the reviewers to provide feedback on our initial submission. The comments were insightful and productive, and we strove to incorporate as many of the proposed changes as possible.

The largest changes to the manuscript are a full uncertainty analysis, new schematic figures, and a discussion of model validation. The uncertainty analysis is described in the Methods and Supplement, but we would like to highlight that we now use results from 540 model runs with varied parameters, as opposed to 18 runs with fixed parameters in the previous submission. While we do not have the ability to increase the size of our IRD dataset at this point in time, we have attempted to clarify several of the concerns about the validity of the field measurements to our process model of sediment entrainment. All reviewers raised concerns about validation, so we have added a new paragraph discussing our model in the context of previously published results. To make space for this, and given several concerns raised in review, we have removed the discussion about geomorphic supply vs. transport-limited regimes. Our new statistical upscaling is more robust than the previous method, and we hope is much more transparent than in the initial submission.

While the specific values have slightly changed, we believe that the overall story and conclusions have remained intact. The previous best estimate of continent-wide IRD fluxes in Overeem et al. (2017) assumed a constant calving front height, uniform sediment concentration, and uniform basal ice thickness. We have used process-based modeling to explicitly simulate debris entrainment, allowing for variable thicknesses in debris-rich ice layers. Further, we have used an extensive, novel field dataset, built from multiple field campaigns in different regions across Greenland, to revise prior assumptions about sediment concentration and more accurately represent the different basal ice facies observed in icebergs in situ. Finally, our statistical upscaling uses a novel relationship that we identified between ice and sediment yield, which has not been reported previously in the literature. While we recognize the inherent variability in the system and the difficulty in upscaling a relatively small selection of sites to the entire ice sheet, we believe that our work represents a rigorous step forward on this problem.

See below for a point-by-point response to the reviewer comments.

Reviewer #1

To quantify iceberg rafted debris in Greenland's total sediment budget, Pierce et al combine field data from 210 iceberg samples across three fjords with a numerical model of erosion and entrainment to estimate IRD fluxes. They find that sediment concentrations span a wide range (0.1–45% by mass), and discuss this in context of frozen fringe sediment and dispersed basal sediment. They find a first-order relationship between ice and sediment yield and estimate that icebergs export ~416 Mt/a of sediment, or roughly one-third of Greenland's total sediment flux. Interestingly, the estimated iceberg sediment flux presented in this manuscript is significantly higher than previous estimates reported by Overeem et al., 2017 and exceeds the values anticipated in the discussions in Andresen et al., 2024. However, I have concerns regarding both the field observations used to constrain the model and the approach taken to validate this elevated flux estimate.

Understood. We will attempt to address these concerns below.

Field observations: Hasholt et al, 2022 describes a method to collect samples of ice from icebergs systematically. They point to their 97% uncertainty range based on only 12 icebergs – and their method is meant for future sampling to provide more realistic values as data collection hopefully increases. They write: We present an estimate of the transport related to calving of ~100 million tons per year with a range from 0.3 to 200 million tons. We are fully aware that 12 samples of icebergs are not sufficient to constrain the transport of sediment reasonably accurate, but we have performed the calculation to illustrate the effects of biased sampling. The simultaneous meltwater-driven suspended sediment fluvial contribution is 21–30 million tons (from table 1 in Overeem et al. 2017). The latter number refers to the meltwater sediment flux from the entire Scoresbysund region. For comparison Overeem et al. 2017 find that iceberg sediment accounts to c. 1% of that contributed from melt water (their table 1) for entire Greenland. While I acknowledge the considerable effort involved in collecting and analyzing data from 210 icebergs, I am not convinced that increasing the sample size alone provides greater confidence in the upscaled estimate. Given the inherent variability in iceberg sediment content and the challenges in achieving unbiased sampling, I believe the current dataset is insufficient to support a robust extrapolation to the ice sheet scale. This is not to discount the value of the fieldwork, which is clearly substantial, but rather to caution that the upscaling appears premature.

These statements from Hasholt et al. (2022) are the driving force behind our entire approach in this manuscript. In that work, the authors engaged in a semi-random sampling campaign, choosing the iceberg “closest to the sample position pre-selected on a map.” Then, “the first sample was taken at a random location where the boat bumped up to the iceberg,” with subsequent samples taken approximately 2-6 m from that initial position. This approach led to considerable variability in the qualitative character of the icebergs sampled: they did not necessarily feature debris-rich basal ice

layers, or if they did, the repeat samples were not guaranteed to come from those layers. Their 97% uncertainty range includes clean, englacial ice, dispersed ice with relatively low sediment, and solid basal ice with very high sediment load. We agree with the assessment that upscaling such a diverse dataset would be premature.

However, our strategy here is different, both in respect to how we gathered field data and our process-based approach to upscaling the results. First, we specifically sampled debris-rich basal ice in this study. We included the data from Hasholt et al. (2022) where field notes, photographs, and sediment analyses could reasonably confirm that the samples were obtained from a debris-rich layer of the iceberg. Therefore, our ice-rafted sediment concentrations *are not* representative of a random iceberg (i.e., we do not measure an “englacial sediment concentration”). They are, however, representative of debris-rich basal ice layers. We are confident that our sampled sediment concentrations from icebergs are well within the range of values observed in situ at glacier margins and in laboratory settings (e.g., Dowdeswell, 1986; Dowdeswell and Dowdeswell, 1989; Andrews et al, 1994; Hubbard et al., 1996 and 2009; Cook et al., 2011; Pierce et al., 2024). Yet, it would not be correct to upscale these sediment concentrations by multiplying against total ice discharge, since the vast majority of the ice column is clean ice, not debris-rich ice.

This leads us to a process-based upscaling strategy. The numerical modeling used in this manuscript follows directly from previous theoretical and experimental work on the thickness and extent of debris-rich basal ice layers. Our goal is to predict the thickness of sediment-rich basal ice layers, such that the total sediment flux can be modeled as the product of thickness, concentration, and sliding velocity across the terminus. As such, the iceberg samples we gathered here are only intended be representative of the modeled basal ice layer, and not of any randomly selected iceberg. For further explanation, Pierce et al. (2024) includes a more thorough discussion of these basal ice facies.

We hope that this discussion also clarified the confusion around Overeem et al. (2017). The reviewer points to Overeem (2017) Table 1, however this refers to ‘englacial’ sediment, defined as sediment within the entire ice thickness. Englacial sediment derives mostly from wind-deposited dust, and concentrations in ice cores are observed to be extremely low (0.001%). However, to estimate glacier basal layer sediment flux in that analysis, the authors assumed a Greenland-wide average calving cliff height (300 m), constant debris layer thickness (3 m), and a sediment concentration of 20% by volume (from sparse field data by Dowdeswell, 1989 and Andrews, 1994), or roughly 50% by mass. In table 1 from Overeem et al. (2017), the 1% value refers to the volume flux of basal ice relative to total calving ice. The basal sediment flux is estimated as 1.92 Gt (Table 1, column 3). Hasholt et al. (2006) estimated a sediment flux associated with Greenland Iceberg calving of 50-500 Mt/yr, based on extremely sparse data. Our model vastly improves on the previous estimate by simulating the processes that form the basal ice layer, rather than making any assumption about its size relative to the calving front. Our current paper also demonstrates that sediment concentration in the basal layer consists of high concentration frozen fringe and low concentration dispersed sediment (while appearing visually dirty throughout the full basal layer in field views of flipped icebergs).

Consequently, our process-based approach sharpens previous estimates, while revising the total amounts substantially downward.

- Andrews, J. T., Milliman, J. D., Jennings, A. E., Rynes, N. & Dwyer, J. (1994). Sediment thicknesses and Holocene glacial marine sedimentation rates in three East Greenland Fjords (ca. 68° N). *J. Geol.* 102, 669–683.
- Cook, S. J., Swift, D. A., Graham, D. J., & Midgley, N. G. (2011). Origin and significance of ‘dispersed facies’ basal ice: Svínafellsjökull, Iceland. *Journal of Glaciology*, 57(204), 710-720.
- Dowdeswell J.A.(1986) The Distribution and Character of Sediments in a Tidewater Glacier, Southern Baffin Island, N.W.T., Canada, *Arctic and Alpine Research*, 18:1,45-56
- Dowdeswell, J.A., Dowdeswell E.K. (1989). Debris in Icebergs and Rates of Glaci-Marine Sedimentation: Observations from Spitsbergen and a Simple Model. *The Journal of Geology* 97, 2, 221-231.
- Hasholt, B, Bobrovitskaya, N., Boge, J., McNamara, J., Mernild, S., Milburn, D., and Walling D. (2006). Sediment transport to the Arctic Ocean and adjoining cold oceans, *Nordic Hydrology Vol 37 No 4–5 pp 413–432*
- Hubbard, B., Sharp, M., & Lawson, W. J. (1996). On the sedimentological character of Alpine basal ice facies. *Annals of Glaciology*, 22, 187-193.
- Hubbard, B., Cook, S., & Coulson, H. (2009). Basal ice facies: a review and unifying approach. *Quaternary Science Reviews*, 28(19-20), 1956-1969.
- Pierce, E., Overeem, I., & Jouvét, G. (2024). Modeling sediment fluxes from debris-rich basal ice layers. *Journal of Geophysical Research: Earth Surface*, 129(10), e2024JF007665.

As a first step, a more informative approach would have included a detailed characterization of the sediment contained within the sampled icebergs—particularly grain size distributions and lithologies—as well as descriptions of sediment banding features, including band thickness, contact relationships, and three-dimensional morphology. This is only partly touched upon – but a systematic report and analysis of the iceberg data from the 210 bergs, including photographs, would be very valuable for our understanding of Greenland’s ‘muddy bergs’. Greenland’s iceberg population is indeed highly heterogeneous, and a deeper understanding of this variability is needed before attempting large-scale quantification of IRD fluxes.

We agree that there is considerable variability in Greenland’s icebergs, and that variability is inherently worthy of further study. We do not agree with the assessment that the rest of the work presented in this manuscript is unworthy of attention until such time as the variability in field observations can be addressed. There are many studies cited throughout the manuscript that use field observations, laboratory experiments, and theory to constrain the processes responsible for building debris-rich basal ice layers. These studies generally agree on a range of sediment concentrations and layer thicknesses resulting from basal sediment entrainment – although again, we agree there is further work to be done. However, modeling a bulk sediment flux does not rely on a lot of the complexity noted here; properties such as grain size distribution, lithology, banding

features, etc. are important for understanding the underlying processes at play, but do not directly control the overall sediment flux.

Validation: The only validation provided in the manuscript is a comparison with sediment flux estimates from Overeem et al., 2017 and Andresen et al., 2024. Pierce et al. argue that the difference between the two studies reveals an unaccounted sediment fraction attributable to icebergs, noting that Overeem reports suspended sediment flux, whereas Andresen reports total sediment flux. However, this interpretation is not correct. Andresen et al. estimate total sediment flux only from marine-terminating glaciers at 1.324 ± 0.791 Gt/yr, while Overeem et al. report two estimates of suspended sediment flux from both marine- and land-terminating glaciers: 0.892 Gt/yr (erosion-based) and 1.28 Gt/yr (meltwater discharge method) (and both studies btw point to a low contribution from iceberg-derived sediment). So, the difference between the two studies clearly cannot be used as a proxy for iceberg sediment flux. A short summary of the two papers, for the readers not deeply acquainted with these studies, would have clarified this.

We appreciate the points raised here, as it appears our exposition of both papers in question was lacking. We agree with this summary of Andresen et al. (2024), although we do not think that the results in that study preclude a larger contribution from iceberg-rafted sediment. In the Sermilik fjord diamicton, they note that the sand content is roughly 20% in cores, whereas sand content in basal ice layers is 50-80%. This still allows for 25-40% of the sediment to be delivered by icebergs, comparable to what we predict here. Of course, this analysis relies on major assumptions about the variability and preservation of grain size distributions in transit. In the plumite, sediment > 63 microns only composes ~1% of the total accumulation. Note, however, that this is not necessarily inconsistent with our approach. We estimate the total sediment contribution from IRD as approximately 1/3 of the sediment reaching fjords, but that is not likely to be true on a fjord-by-fjord basis. Sermilik fjord's outlet glaciers have a moderate ice yield (due to very large catchment areas) and high runoff rates (as is typical of southeast glaciers). So, it is entirely reasonable that the IRD fraction is much lower than the meltwater fraction in this setting. Also, as they note, the Andresen et al. study does not have access to cores within appx. 30 km of the calving front. Given that the basal ice layers with the highest sediment concentrations are located near the bed, we would expect those layers to melt out earliest, and thus closest to the calving front. We agree with this review that this is an important discrepancy to address directly in our manuscript.

Regarding Overeem et al. (2017), there may be additional clarifications required. First, the upscaling approach did focus on meltwater-driven sediment transport in both marine-terminating and land-terminating glaciers, as stated here. But, it is important to note that sediment from land-terminating glaciers ends up in proglacial rivers, which drain into fjords. Thus, the vast majority of sediment from these two pathways of transport (meltwater and icebergs) end up at the same sink. Further, Overeem et al. (2017) do provide two different estimates of suspended sediment flux, but actually point to a massive contribution from icebergs of 1.92 Gt per year (as opposed to the relatively low

contribution of ~ 14 Mt/yr from englacial sediment). We describe the estimation methodology above, but clearly the error in that approach lay in assuming that the entire debris-rich ice layer of 3 meters had a uniform sediment concentration around 50% by mass. The current manuscript revises this estimate down by more than half, based on a much more thorough understanding of the typical ranges of sediment content in basal ice layers.

Despite these clarifications, we do understand the overall point here: it is not fair to use the difference between estimates in Andresen et al. (2024) and Overeem et al. (2017) as a validation step. We have clarified throughout the manuscript that this is a discussion point – that is, a potential consequence of our approach, rather than a piece of evidence in its favor. We have also added a paragraph discussing more local validation steps where possible.

Moreover, the methodologies in these studies differ significantly. Andresen et al. rely on sediment core accumulation rates, while Overeem et al. use remote sensing and modeling of meltwater fluxes. Both approaches involve large uncertainties, which are acknowledged in their respective discussions in these papers. However, the broad agreement in their flux estimates, despite such different methodologies, suggest they are both on the right track. But their differences reflect these methodological differences and limitations—not a missing ~ 430 Mt/yr of sediment. For example, Overeem’s method is limited by sparse data near deep glacier termini, which alone could account for substantial underestimation. Likewise, Andresen study is limited by not having sediment cores right next to the glacier margins. The manuscript’s interpretation of this discrepancy as iceberg-derived sediment is therefore problematic, especially as it forms the sole basis for validating the model’s sediment yield.

It is true that the methodologies in both studies differ, but it is also important to note that the studies are not measuring the same components of the system. Andresen et al. use sediment core accumulation rates, so their upscaled result should be viewed as the total sum of sediment that reaches fjords. Overeem et al. use a remote sensing method to connect meltwater fluxes with suspended sediment load, and thus are only able to predict the contribution from suspended sediment (not bedload, nor iceberg sediment). It would be odd if the studies were in close agreement, since sediment does not exclusively reach fjords through suspension in meltwater. Note that we do not include Overeem et al.’s assessment of ice-rafted debris fluxes or englacial sediment fluxes here. We do elect to use the upscaling relationship between glacial erosion and sediment flux from Overeem et al. with a stronger fit to the field data, as that was the value the authors themselves chose to elevate throughout the manuscript, but we made that decision more transparent in the current manuscript.

A question that comes to mind here is: Why have they not compared with sediment core data from Greenland’s fjords and shelf presenting iceberg rafted debris fluxes?

We feel this comparison is out of scope of this study. There is an inherent problem in using sediment core data to validate bulk fluxes, as noted by the discussion brought up earlier from Andresen et al. (2024). Sedimentologists identify IRD as coarse, angular sediment grains in marine cores that would have fallen out of suspension in a plume much closer to the terminus. However, a substantial proportion of IRD derived sediment (anywhere from 20-50%, depending on the study) is fine enough to be indecipherable from fluvial suspended sediment (f.e. Smith and Andrews, 2000). And, to further complicate matters, grains may be worked by the subglacial fluvial system before becoming entrained in basal ice, such that angularity is not a precise identifier either. Most studies of IRD in cores use sedimentological evidence – e.g., magnetic susceptibility – to identify phases of elevated/diminished IRD fluxes. However, this does not always translate neatly back into a flux in mass/time or even mass/area/time (the second of which has an inherent problem with deposition location – see the points about Andresen et al., 2024, above). Despite these reservations, we have added a comparison to a few different validation approaches that may be appropriate for this work.

Smith, L. M., & Andrews, J. T. (2000). Sediment characteristics in iceberg dominated fjords, Kangerlussuaq region, East Greenland. *Sedimentary Geology*, 130(1-2), 11-25.

Some minor comments

Line 33 and 34: Some of these studies also point to plume sediment as a shoal-builder, in particular the studies of temperate Alaskan glaciers.

Good point. The sentence is now: “Within the grounding zone of glacial outlets, sediment deposits contribute to shoal formation, inhibiting tidewater glacier retreat.”

Line 36: heightened levels of chlorophyll are linked with sediment in the Southern Ocean where iron is a limited nutrient. Around Greenlandic icebergs local upwelling of macronutrients from Atlantic-sourced water may be the cause of heightened chlorophyll, and not the elements contained in the sediment. So, it would be correct to mention these observations goes for the Southern Ocean.

Thank you for identifying the flawed statement here. We have updated it in the revised draft.

Line 113: what are the characteristic sediment grain size distribution in the frozen fringe sediment, and in the dispersed basal ice facies according to the referenced literature? And are these based on in situ observations by landterminating margins? This information is highly relevant when linking the sediment sampled from icebergs with the sediment entrained underneath the glacier. Do the grain size distributions match?

We only have limited quantitative grain size analysis for most of the IRD samples, and none from most of the till that underlies the Greenland ice sheet. The grain size distribution in the frozen fringe is inherited directly from the grain size distribution in the till beneath the glacier (this is a fundamental part of the theory on frozen fringe mechanics). Particle migration into dispersed basal ice, above the frozen fringe, is an open question in glaciology. We have used the model from Pierce

et al. (2024) in this work, as we believe that model represents the highest fidelity to real-world observations from glaciated margins.

Line 154. What is the role of subglacial meltwater derived from surface melt? Does channelization of meltwater near the glacier margin influence the sediment content at the icebed interface? This likely represents an important process through which basal sediment is transferred into the meltwater plume. If this mechanism is not accounted for, estimates of sediment yield from icebergs may be significantly overestimated. While these processes are reported on in the methodology section is not mentioned in the discussion and it is not clear to me how IRD and plume sediment is partitioned in this study.

We agree that subglacial fluvial transport is likely a dominant sediment source, especially in many of the outlets in southwest Greenland. We do not explicitly simulate subglacial fluvial transport here, nor do we simulate meltwater plumes. This model is focused on basal sediment entrainment, although we do agree that the interactions with the fluvial system are certainly interesting and worthy of further study. We make the case here that frozen fringe is active beneath a vastly greater area of the ice sheet than channelized meltwater, such that these interactions would be limited to a relatively small area of the bed. Importantly, there is sufficient erosion in our model to allow for sediment transport by both pathways – that is, we would not expect our IRD fluxes to be greatly reduced by adding a fluvial component to the model. However, modeling subglacial sediment transport by meltwater is itself an open question in glaciology, such that a fully coupled model of the system would be a vision, but far out of scope for this work.

Line 156-157: How do the erosion rates compare to previous reported estimates of erosion rates from Greenland cathments? I believe the ones reported by Pierce et al. are 100-fold higher than previous estimates?

Good catch. The x-axis label was incorrect on this plot. We have elected to remove it, as it raised significant confusion among all reviewers and did not garner much positive feedback. Our peak erosion rates are that high, but only at limited points in space. The catchment-averaged erosion rates are on the order of 0.01-0.1 mm per year.

Line 165: is this not expected when the same surface velocity and terminus width are primary variables in calculations of both sediment yield and ice yield (data in Mankoff)? The derivation of ice yield is reporting in the methodology section, but there is no reference to this on the Figure 4C.

In our understanding, an appropriate null hypothesis would be that sediment discharge scales with ice discharge (e.g., Methods section of Overeem et al., 2017). And as pointed out here, that is precisely because surface velocity and terminus width are used in both calculations. But, that is not what we find here. Instead, we find that the ice yield (surface velocity x terminus width x calving front height) scales the sediment yield (sliding velocity x terminus width x debris-rich ice thickness). While surface and sliding velocity are related, they are not equal to one another for many of these

systems. And the thickness of debris-rich ice layers (our primary model output) is fully independent of any other variables used in the ice discharge or yield calculation.

Line 245-248: I suggest to delete this statement. Sea level will decrease by Greenland's glacier due to isostatic rebound and reduced gravitational pull; it will not increase here. The question is if shoaling can prevent incoming warm Atlantic waters and 50 cm of sediment per decade would do nothing here.

Good catch, thank you for pointing out the flawed logic. We have exchanged that sentence for a comparison with a grounding zone wedge identified at Whillans Ice Stream.

Line 228: grain sizes and angularity....would be great to be more specific here.

Added more specific descriptions.

Line 230: how is that known?

We are not sure what this statement is referring to. We know that prior to iceberg calving, dispersed basal ice melts out later than frozen fringe because it is located higher in the ice column. While calving and subsequent mixing in mélange can stochastically alter iceberg orientation, it is very unlikely to find a geometry where dispersed basal ice melts out earlier than the clean ice or frozen fringe on either side of it.

Line 232-234: to be clear to a broader audience – specify what is meant by reconstructions.

But be also aware that reconstructions of iceberg production and dispersal changes in the past are based on the coarse fraction (for the reasons mentioned by Pierce et al.).

Clarified further in the text.

Line 235-237: I agree, and for this reason grain size distributions of the samples would have been informative.

Unfortunately, we do not have quantitative grain size distribution or chemistry measurements for all of the IRD samples, which is why we omitted this characteristic from the discussions. This is a good point, and will have to wait for future work. For the current paper, we chose to focus on the physical properties that are most influential for bulk sediment fluxes.

Line 250-252: it says: "This assumption holds if the system is limited by supply- that is, if the glacier has the capacity to transport all available sediment." Should it not be "is not limited by supply"?

Apologies for the confusing wording here. If the system is limited by supply, this implies that all available sediment is moved. Otherwise, it would be limited by the glacier's ability to transport the sediment. Regardless, this paragraph is not a critical part of the discussion, and has been removed to make space for more relevant topics raised earlier in this review.

Reviewer #2

Review of "Sediment transport by Greenland's icebergs" by Pierce et al.

In this paper, the authors quantify the volume of sediment exported by icebergs calved from Greenland's glaciers. They focus on three glacial systems and extrapolate their findings to the entire ice sheet. Their interpretation is based on numerical modeling that encapsulates current understanding of sediment storage in glaciers and associated erosion/transport processes. The main finding is that approximately one-third of sediment export from Greenland occurs via icebergs. This represents an important contribution, and I hope to see it published.

Thank you!

That said, I have a few recommendations which I hope will help the authors clarify certain points:

1) While I understand the different components of the numerical model, it took me some time to fully grasp its structure. A schematic illustrating the model's elements would be very helpful. This is important because the model underpins much of the interpretation—it includes calculations of sediment flux, transport by frozen fringes, and dispersion, which are all clearly presented. However, the mechanism by which erosion feeds sediment into the ice layers remains unclear to me, even though equations 4 and 5 incorporate erosion as a source term. As a result, the discussion on supply-limited vs. transport-limited regimes is somewhat unclear.

Good to know that this is a point of confusion. We have added a schematic in the Methods to help guide readers, and have written more carefully about this in the main text.

To clarify directly here: the frozen fringe component of the model is responsible for entraining sediment into basal ice layers. The erosion component determines how much sediment is available – we have many regions where the fringe theory predicts 1 cm of sediment entrainment over some amount of time, but erosion can only provide 5 mm, for example. In that case, the fringe would only entrain the available 5 mm.

2) The authors use a single set of values for the erosion parameters. While I agree with the decision to keep the erosion model simple, I would be interested to see how sensitive the results are to the choice of K_g and l . Does this affect the results shown in Figure 4a—for example, the ratio of erosion to transport? Additionally, how much sediment could accumulate if erosion continues to outpace transport over centuries, millennia, or even tens of thousands of years?

Good question. We have added K_g and l to the newly implemented uncertainty quantification to help answer this. Figure 4a was included to help justify the choice of a simple erosion model. Now that we are using a range of values for K_g and l , we decided to remove Figure 4a and the discussion of supply vs. transport limited regimes. Without a fluvial transport component, we do not think we have the ability to answer the last question sufficiently – which is another justification for removing

the discussion on supply/transport from the revised manuscript. The answer to this question also depends quite a bit on climate variability that we do not model here, since our sliding velocities are derived from surface velocity composites from 2023 (in the updated version of the manuscript). We are poised to explore this question on long-term variability in future work.

3) The authors show that sediment content in icebergs follows a long-tail distribution. If I understand correctly, this is based on the assumption that the icebergs are representative of the basal layer. But are we certain that icebergs aren't just fragments of the ice front? In other words, could they be breaking through the middle of the sediment layer? I would expect such a process to also produce a long-tail distribution. How can we be confident this is not the case? This comment might be re-stating the same view of the problem that we take in the manuscript. We do not sample icebergs at random (c.f., the discussion above about Hasholt et al., 2022). Instead, we sample icebergs that have exposed sediment-rich ice layers to the surface (as best identified by basal ice facies descriptions in the field, see Hubbard et al., 2009). It is certainly possible that icebergs are breaking through the middle of the sediment layer, or otherwise sampling from it at semi-random, but this would only produce a long-tailed distribution in sediment concentrations if there is a gradient in sediment concentrations throughout that layer (because we are not including clean iceberg samples in our analysis here). This is a possible process explanation for what we observe. We should also consider the stochastic nature of sampling from icebergs. Because the enriched frozen fringe has the opportunity to melt out as soon as basal ice is exposed to warm ocean water, often before the calving front, much more dispersed basal ice survives into the fjord than frozen fringe. This also produces a long-tailed distribution in what is available in the fjord for sampling. Regardless of process, the long-tail distribution in sediment content should be evidence for the fact that enriched sediment layers sit beneath dispersed sediment layers. This is not necessarily new information, but does reinforce what we expect from field studies at glaciated margins.

We have added a new schematic figure to the manuscript to help clarify this point.

Hubbard, B., Cook, S., & Coulson, H. (2009). Basal ice facies: a review and unifying approach. *Quaternary Science Reviews*, 28(19-20), 1956-1969.

4) The authors present sediment export in terms of volume. One of the most striking results is the correlation shown in Figure 4c between ice yield and sediment yield. I wonder what the relationship would look like if plotting ice velocity vs. sediment yield instead?

Here is a plot with maximum sliding velocity vs. sediment yield (just using base parameters, not the median of UQ runs) for each catchment (in log-log space). Almost all of the outlet glaciers here end up with terminus velocities > 1 km/yr, which is fast, but not atypical for Greenland. The greater velocities do not necessarily trend towards greater sediment delivery, because they are often accompanied by greater rates of basal melt, which reduces the frozen fringe thickness.

Specific comments:

Abstract, line 28 – The paper doesn't address how sediment transport may evolve (or has evolved) in response to climate change. That said, I would be very interested to read any projections or hypotheses the authors might offer. Will we see more sediment delivery or greater sediment storage within glaciers?

The final paragraph of our discussion provides our best perspective on this question, based on the results in this manuscript.

Line 42 – It would be helpful to also indicate the total sediment volume here.

Line 73 – The error bar is missing after "416 Mt."

Good catch, thank you.

Lines 90–93 – How can we be certain the sampling captures the entire basal layer rather than just part of it?

Discussed in a bit more detail above – we are almost certain that most sampling does not capture the entire basal layer. As can be seen from Figure 2, only a limited number of icebergs were sampled directly near the calving front. This is both due to safety concerns as well as due to inaccessibility of the mélange to our vessels. If it did, we would expect a much greater prevalence of frozen fringe, given that our model predicts fringe thicknesses on the order of tens of cm to meters. We have clarified this further in the text.

Line 157 – These erosion rates are quite high. How sensitive are they to the model parameters?

Good catch. There was an error in the plot axis labels. The erosion parameters have now been wrapped into the full uncertainty quantification, so variability in erosion rates is included in the overall envelope of model outcomes. To answer the question directly, the erosion rates are very sensitive to the choice of Kg and l , especially l , considering the range of sliding velocities we have here (see figure above).

Figure 3 – What causes the patchy appearance in the data?

The patchy appearance in the data is fundamentally caused by a patchy sliding velocity field. Where there is no sliding (e.g., modeled deformation matches or exceeds measured surface velocity), we do not allow erosion, which thus precludes any sediment entrainment. Where there is a small amount of sliding, eroded sediment is quickly entrained in frozen fringe, where it can then migrate into dispersed layers above. But, because there is negligible sliding in the intervening regions, very little entrained sediment gets advected into neighboring grid cells.

Line 213 – Is this really a controlled setting?

Not at all. We are referring to previous work on basal ice-sediment interactions.

Lines 259–260 – I suppose this depends on the model used?

True, but we will save that discussion for future work. As discussed above, we think this paragraph detracts from the overall focus of the manuscript.

Line 273 – What is the expected behavior of erosion over time?

Our understanding is that erosion should increase with increasing sliding velocities in the short term (the next several decades or so), but stabilize or decrease in the long term. We have updated the discussion to reflect this.

Line 363 – It's unclear how erosion modifies the till layer. Some clarification would help.

Clarified in the text.

Reviewer #3

Pierce et al. present modelled estimates of ice-rafted debris (IRD) in three fjord systems. Model outputs from 11 glacier basins are used to derive an empirical relationship between iceberg sediment yield and an existing estimate of grounding line discharge. This relationship is then used to estimate the total sediment yield from Greenland's solid ice export. This result represents the 'missing piece' in Greenland's total sediment production, so the result is important. Reflective of this importance, I anticipate that the community will readily adopt the values for IRD presented in the manuscript. In anticipation of this, it is essential that the assumptions used to derive those values are clearly presented, the sensitivity of the modelling to various parameter choices are comprehensively quantified and model validation is performed wherever possible. In its present form, the manuscript could (and should!) do all of these better, which leaves the reader lots of reasons to be sceptical of the presented IRD values. My major comments below describe where I think the manuscript falls short in these respects, and provide some suggestions of how to make the result as robust, transparent and convincing as possible.

Thank you for acknowledging the importance of the work and for pushing us to improve the transparency and rigor of the analysis. We will address the specific concerns as they appear below. Overall, this was an exceptionally thorough and helpful review, and we appreciate the time that went into it.

Major comments (in no particular order)

1) There are many parameters in your model for which the appropriate value is unknown or poorly constrained, yet no sensitivity tests are presented to examine the influence of parameter value choices on modelled sediment thickness. Related to this, one value for dispersed sediment concentration is used in the simulations, yet visual examination of Figure 2 suggests significant differences between fjords – please quantify this difference and justify the use of a fixed value. The manuscript even states on lines 95-97 that sediment concentrations vary significantly among nearby samples and that this variability needs to be accounted for in models, yet a fixed value is still used and the impact of this choice on the results is not presented. Please provide a thorough sensitivity analysis of all uncertain parameter values and quantify how they affect the modelled IRD in each of the fjords with field data, as well as the reconstructed IRD for all of Greenland.

This comment is asking for a tremendous amount of work. We have done our best to meet this standard, but a full sensitivity analysis is far beyond the scope of this manuscript. The core components of the model are shared with Pierce et al. (2024), which does present a parameter-by-parameter sensitivity analysis for the sediment entrainment component. But, we think the most important point to address is the variation in model results based on variations in the uncertain parameters. That is, what is the change in our predicted IRD fluxes given a reasonable change in model parameters? To address this, we have undertaken a thorough uncertainty quantification, focusing on the points raised by the last two lines in this review comment. Informed by sensitivity experiments in Pierce et al. (2024), we performed a Monte Carlo uncertainty analysis on the nine most impactful variables, varying each across a very large range of plausible values (see Supplement). Importantly, we made sure to include variables with strong influence over each of the key input fields to the model, particularly effective pressure and sliding velocity, as well as the core components of the model, including erosion and sediment entrainment. We ended up performing 30 model runs for each catchment. This sample size stretched our computational limits, and did require significant model refactoring to run in a reasonable amount of time. We have added a new section to the Methods describing the protocol. (Revised) Figure 5 now includes distributions for modeled sediment yield from every catchment. All error bars in the manuscript are now derived from the set of all 540 model runs from that Monte Carlo scheme.

As a side note, there are not really significant differences in dispersed sediment concentrations between the three fjord systems. The median values are within 1% of each other, and the interquartile range is comparable, with a higher upper end for Ikerasak than the other two fjords (about 12%). The higher values at Ikerasak are a result of more samples taken closer to the margin,

simply due to our ability to access different parts of the fjord. Our conceptual model (which is reinforced by in situ samples at glaciated margins from prior studies) indicates that higher sediment concentrations should be found nearer to the base of the glacier. Because the lowermost ice layers melt out earliest, with the exception of stochastic overturning of icebergs, this means that the closer samples from Ikerasak should have generally higher concentrations. Importantly, our field observations and analyses of basal ice facies do not suggest that the overall population of samples from Ikerasak were systematically different from the other two fjords.

Pierce, E., Overeem, I., & Jouvét, G. (2024). Modeling sediment fluxes from debris-rich basal ice layers. *Journal of Geophysical Research: Earth Surface*, 129(10), e2024JF007665.

2) As far as I can tell, your simulations and scaling up assume that there is porous till everywhere in all of Greenland's tidewater glacier basins. This assumption is not clearly stated and the impact of the assumption is not quantified. There are very few constraints on the characteristics of the substrate in tidewater glacier basins, but boreholes in land-terminating catchments indicate extensive areas of bedrock (<https://agupubs.onlinelibrary.wiley.com/doi/full/10.1002/2017JF004201>). Given this uncertainty, it seems unreasonable to assume the extreme end-member that there is till everywhere. This is clearly an important assumption because the frozen fringe appears to provide the vast majority of sediment transport in your model (Fig. 4). Please either provide convincing justification for this choice or modify your assumptions in the simulations based on the little available evidence in the literature. Whichever you choose, please state your assumption clearly and, if reasonably practicable, quantify the impact of that choice on your results.

Our simulations and upscaling do not assume there is porous till everywhere. We absolutely agree with this comment that there should be extensive bedrock regions beneath the ice sheet. Two points: first, our erosion component sets the till availability for the rest of the model – the frozen fringe component only entrains sediment up to the maximum amount available. Second, sediment entrainment in basal ice actually provides a mechanism for these extensive bedrock regions observed beneath Greenland. If frozen fringe entrainment outpaces erosion rates, there can be entrained sediment in the basal ice of the glacier *and* a bare bedrock surface beneath it. In fact, because frozen fringe is such a (spatially) widespread process, this outcome is very likely throughout the interior regions of the ice sheet.

As an action item, we have clarified the first point throughout the text. Similar comments were raised by other reviewers, so it is clear that our explanation of the relationship between modeled erosion and frozen fringe growth was lacking.

3) As presented, the only model validation is the Greenland-wide comparison with the residual IRD in Overeem et al. (2017), which in turn relies on scaled-up modelled ice sheet runoff (which is known to substantially misdiagnose measured runoff) from both the GrIS and surrounding ice caps, and, if I understand it correctly, also does not account for IRD that escapes from fjords in

icebergs. The presented validation does not remove the effect of sediment export from surrounding ice caps in the Overeem et al. (2017) estimate, which would clearly affect the residual IRD value. I appreciate that there is very little data available for validation, but could you also use the Overeem et al. (2017) estimate on a catchment-by-catchment basis and compare those with your simulated sediment export? Could you directly compare your simulated sediment layer thickness at the terminus with those observed by you and others in the field? (Related to this, it would be helpful if your modelled sediment layer thicknesses were presented clearly – see comment below). If the sediment concentration evolves within the model, could those values be compared to your measured values? It's not clear that the model is internally consistent: do the modelled erosion and transport rates combine to provide the observed sediment concentration values? In addition, could you simulate an additional glacier basin (or ideally several basins) and compare the resulting modelled IRD to the reconstructed IRD based on grounding line discharge? I appreciate that considerable effort is required to create new model domains and a small amount of computational time (a few hours) is required to run the model, but this would make the upscaling more convincing.

We agree with this point, and have added a new paragraph on different model validation steps. Based on other reviewer feedback, we have reframed the comparison with Andresen et al. (2024) and Overeem et al. (2017) as a discussion point, rather than a “validation” step. In our (revised) Figure 6, we have added a regional comparison with fluvial fluxes from Overeem et al. (2017). We have added comparisons with field observations based on the feedback here. The sediment concentration does not evolve in the model, but is reasonably well-constrained by the field measurements.

We have added a new schematic to the Methods illustrating how the model operates. It is not entirely clear what “internally consistent” means in this context, but the model conserves mass, if that is the concern here. All eroded sediment either sits in the till layer or is added to frozen fringe, and all dispersed sediment is taken from the underlying frozen fringe. Basal melt removes sediment from frozen fringe, or the dispersed layer if no fringe is present, and adds that sediment back to the till layer. We have clarified this throughout the manuscript.

Given the amount of additional computational time needed for uncertainty quantification, as well as the constraints on manuscript length and the number of display items, we have elected not to simulate additional catchments for validation. While a successful model run takes a small amount of time, as noted above, model runs that fail due to domain geometry, flawed input data, or numerical issues from linear solvers take considerably more time to debug. And, our time-per-catchment has increased by 30x due to the uncertainty quantification step. However, if the upscaling is still not convincing after our uncertainty quantification and additional discussion of validation steps, we are willing to revisit this idea.

4) The reconstruction of Greenland-wide IRD could be presented more transparently. Firstly, please provide details regarding the estimation of the uncertainty bounds on the fit – this

uncertainty should be updated to reflect the sensitivity analysis recommended above.

Secondly, please provide the RMSE (or a similar metric) for the performance of the fit against the modelled basin IRD fluxes. Third, please describe the distribution of ice yield amongst the unmodelled catchment within the bounds of the fit – i.e. for how many basins do you need to extrapolate the fit? What fraction of the total IRD do they provide? Over what time period is grounding line discharge averaged to produce the fit and reconstruction and how does that differ from the average discharge during the period over which the ITS_LIVE 120 x 120 m ice surface velocity was generated?

These are all great points, and we have added a new Methods section to describe the upscaling in more detail. The uncertainty bounds on the fit have been defined clearly. The RMSE is now included in the figure and in the text with additional context. The distribution of ice yield has been defined in the text (80% of Greenland's outlet glaciers fall within the bounds of our fit, representing 76% of the ice sheet's total solid discharge). The time interval has been clarified in the text.

By producing the fit as presented, you are implicitly assuming that your simulations capture all of the variability of sediment layer thickness around Greenland and that all of the variability in total sediment yield is due to spatial variations in sliding speed. You should back this up with your model results – you have modelled sliding speed and modelled sediment layer thickness, so use those to demonstrate that sediment yield mostly varies with sliding speed not sediment layer thickness and show that sediment layer thickness at the terminus does not have a consequential range. You have presented an estimate of Greenland-wide IRD, and in many ways this is the value everyone wants to see, so either make it convincing or make it clear that it would not be prudent to attempt it because the uncertainties are so large.

We do not fully understand this comment. Why is it the case that the variability in sediment yield can only be attributed to spatial variations in sliding speed? Sediment flux is the product of layer thickness, terminus width, sliding velocity, and sediment concentration; yield is this value divided by catchment area. Sediment layer thicknesses vary quite a bit, due to the nonlinear interactions between effective pressure and basal melt rate that set frozen fringe growth, plus the spatial pattern of advection that governs dynamic thickening/thinning of sediment layers. However, the key variables that control sediment layer thickness (basal melt and pressure) are implicitly related to sliding velocity, even if the exact nature of that relationship is not well constrained. Ice yield is set by surface velocity, terminus geometry, and catchment size. So, the variability in the extrapolation is due to the relationship between sediment layer thickness, terminus height, and differences between surface and sliding velocity. This allows for considerable variability caused by basal processes, such as basal melt and effective pressure, which will not be directly explained by ice yield, but are certainly related to ice discharge.

In more quantitative terms: our fit between ice and sediment yield has an extremely low p-value, indicating that the relationship is significant (at the 99.999% confidence level). However, the R^2 value shows that ice yield only explains 44% of the variability in the sediment transport system. This follows from our discussion above about the inherent nonlinearities in the system (and the variability

introduced by modifying model parameters in the uncertainty quantification scheme, none of which change the ice yield). However, the RMSE shows that the predicted fluxes are reasonable at the catchment scale, and the confidence interval around the extrapolated fit reflects all of the variability discussed here.

We think that our updated results show that the uncertainties are not “too large” – we leave the question of whether or not this is convincing to the reviewers.

5) Data availability: the model output and reconstructed IRD estimates for each glacier basin, glacier region, and summed across the ice sheet (as shown in Figure 5), should also be made available. These are the primary outputs from this manuscript.

Understood. We have included these outputs in csv files.

Minor comments

Line 25: Please clarify what the values in parenthesis represent. Or it may be clearer to just provide the range initially, then the most likely value.

Agreed, this has been updated.

Line 31: “Originating far inland” is, I think, inaccurate. Does IRD not accumulate up to the terminus?

Removed “far.”

Line 35: This is true for Antarctic icebergs – indeed, each of the references given here are of studies focusing on the Southern Ocean. The subject here is Greenland, where iron is less frequently a limiting nutrient and any increase in biological activity around icebergs may be due to upwelling of nutrient-rich Atlantic Water into the photic zone. You should either clarify that this sentence refers to Antarctic icebergs or modify the sentence to reflect the Greenland/North Atlantic literature.

Modified, good point.

Lines 41-43: these two values conflate the results of two studies. I suggest you modify the first sentence to present the total sediment fluxes from both Andresen et al. (2024) and Overeem et al. (2017), and make it clear that the runoff-only value is based only on the results of Overeem et al. (2017).

This paragraph has been edited to distinguish between the two studies.

Line 141: “cm a-1” isn’t an accurate representation of speeds in the majority of the ice sheet interior. The ITS_LIVE 120 x 120 m mosaic suggests most areas in the interior have speeds >10 m a-1 and GNSS measurements 140 km from the margin showed speeds of ~50 m a-1.

Is this comment referring to surface velocity or sliding velocity? Measured surface velocities in these regions are dominated by ice deformation, so there is relatively little contribution from sliding. We

have changed the parenthetical to refer to less than 1 m/yr, but sliding velocity is certainly not 10 m/yr or even 50 m/yr through much of the interior.

Line 152/153: Can you back up this statement about the frozen fringe with numbers? This is a key assumption in section 4, so I think it is important that it is quantified.

We have quantified the model area where (1) effective pressure is great enough to allow ice to infiltrate sediment and form a frozen fringe, (2) erosion rates are non-zero, and (3) basal melt rates indicate the ice sheet is not frozen to the bed.

Line 156: The use of “maximum” here is confusing. It would be clearer to just say that the highest catchment-average erosion rates are X and Y at Eqip Sermia... etc. Those two erosion rates also seem unrealistically fast.

Based on feedback from all reviewers we have removed this section.

Lines 171-173: The meaning of this sentence is a little hard to figure out, or how you determined that the model is likely underestimating the true IRD export. What do you mean by “choice of exactly where to measure sediment fluxes”? Since your model is in steady-state, do you not just integrate the transport rate? If you have to specify some model grid cells near the terminus, then perhaps you can be more specific than just “highly sensitive” e.g. by what proportion does it change if you sample one pixel upstream?

This was due to a mismatch in the input data. We have gone through and corrected those catchments by hand, ensuring that we draw the sediment “flux gate” at the most likely terminus location based on visual satellite imagery and updated ice geometry data products. This only changed the flux values for 4 catchments, and allowed us to pull all catchments from our field sites into the overall fit.

Line 272/273: I don't think your results justify this statement. The statement implies that (1) outlet glaciers will continue to accelerate in future; (2) as outlet glaciers accelerate, erosion rates will increase; (3) that your relationship between ice yield and sediment yield is robust at a single accelerating glacier over time, rather than only across multiple glaciers which are assumed to be in steady-state, and that the increased sediment production is greater than the uncertainty in the fit; (4) even though the system is transport limited, increased erosion will also lead to more transport.

We agree that this statement overstated our conclusions.

Figure 4c: I recommend changing the units on this figure to Gt or Mt per year, rather than providing them per unit area. The metric of interest here is the export of sediment across the calving front, so it does not make sense to provide it per unit area.

We strongly disagree with this point. While the physical quantity of interest is mass transport across the calving front, Figure 4c (now Figure 5) is showing a relationship in yield space that does not

appear in flux space. The transformation from flux to yield is the central idea of this figure, and one of the most important ideas in the manuscript more broadly.

Figure 4c: Are there points missing from the Kangertittivaq region on this graph? I am assuming that each point corresponds to one of the labelled outlets in Figure 3 – there are 11 labelled outlets but only 8 points on the graph.

The smallest catchments (in terms of ice yield) were omitted in the initial draft of this figure. However, the updated figure (now Figure 5) has all catchments from our field sites.

Reviewed by: Benjamin Davison

Reviewer #3 (Remarks on code availability):

I have not reviewed the code. From a quick look at the zenodo repository, it looks like the results would be possible to reproduce with considerable effort. I could not see the model output in the repository, which seems like a major omission. As per my major comments above, please provide the model fields from the target basins, the IRD export from those basins as provided by the full model, and provide the reconstructed IRD for all basins, all ice sheet regions, and the ice sheet as a whole.

We have added model output and some additional guidance to the repository.

Manuscript NCOMMS-25-17411B

Response to Reviewers

Brief plain-language summary:

Greenland's icebergs transport ~450 megatonnes of sediment to its fjords each year, representing one-third of the ice sheet's total sediment export. This ice-rafted debris builds shoals at tidewater glacier margins and provides key nutrients for marine ecosystems.

Reviewer #1

All minor changes not noted here were accepted.

This is a highly cited paper, so it would have been helpful to emphasize that Greenland's total sediment export is higher when the basal contribution is included. In the current manuscript (line 71), the 1.92 Gt yr⁻¹ value is referenced, but the methods section in Overeem et al. reports 2.88 Gt yr⁻¹. The current study certainly doesn't stand or fall on this point, but it is confusing when table values do not match the referenced calculations.

Based on this comment, we have elected to use the 2.88 Gt/yr value throughout this manuscript. As noted, the current work does not stand or fall on this point, so we agree that it is best to remain consistent.

I suggest clarifying around line 47, to avoid confusion, that Andresen et al. (2024) estimate sediment export only from marine-terminating glaciers to fjords, to make the scope distinction explicit and prevent misinterpretation of the comparison (or those comparisons coming later).

Agreed, we have accepted the suggestions for lines 44-50.

Line 205-208: the authors write: "Despite these caveats, sediment fluxes from IRD and meltwater fall within the range of uncertainty of fjord accumulation rates, suggesting that these are the two most significant pathways of sediment transport from the Greenland ice sheet." I am not sure what other significant pathways of sediment transport there are than icebergs and meltwater.

Other pathways including mass wasting events (e.g., landslides, rockfall, etc.), turbidity currents, wind-blown sediment, till deformation, and so forth. We agree with this reviewer that those other pathways are not likely to be significant, but this study presents a much more quantitative argument for that point, hence the inclusion of this line.

With Overeem et al. (2017) estimating total sediment flux from both land- and marine-terminating glaciers (excluding basal IRD), Andresen et al. (2024) combining meltwater and IRD fluxes from only marine-terminating glaciers, and now Pierce et al. focusing only on IRD, the community has three complementary and independently validated estimates of Greenland's sediment export. Why not summarize this in the manuscript? It is entirely appropriate to compare these studies to assess how their results overlap, and a clear outline can avoid confusion about which flux components are included in each estimate.

We agree with this assessment, and hope that the current manuscript does exactly this. In the introduction, we outline each of the previous studies, summarize their methods, and present their main result. Then, in the inset in Figure 5, we include three bar plots for a visual comparison of the three studies. And in the discussion, we take a paragraph to compare the different methods and the main reasons why they differ.

Line 71: the authors write: "Not only is this estimate high compared to reconstructed accumulation rates in fjords ... etc." - a reference is needed for this statement.

We have added a reference to Andresen et al. (2024) for this, and clarified the text to reflect that. The purpose of this statement is simply to point out that an additional 2-3 Gt of sediment per year would be difficult to reconcile with the previous result of 1.3 Gt per year reaching Greenland's fjords.

The authors also write (Line 297-299): "We are observing increased mass loss from Greenland's tidewater glaciers as ocean warming contributes to enhanced rates of iceberg calving, increasing the ice sheet's overall ice yield [39-41]." There is a wealth of paleoclimate/IRD studies that shows this.

We added the reference offered by this reviewer, and updated the text as per another reviewer's comment.

I was referring to the statement: "The dispersed layer also contributes to the total amount of IRD exported in a catchment, but is dominated by fine sediment with a much lower proportion of large particles." How do the authors know this, given that grain size composition was not measured? A reference to theoretical or observational work supporting this claim would strengthen the statement.

While we do not have comprehensive grain size distributions for our samples (as discussed in the prior round of reviews), we did look at a few distributions to get a limited understanding of this point. Additionally, our field notes included qualitative estimates of D_{90} , which are consistent with this statement. This is also consistent with our understanding of pre-melting dynamics and regelation (the underlying theory on which this model is built).

Reviewer #2

There were no further comments from this reviewer.

Reviewer #3

Title: consider "Sediment transport by Greenland's tidewater glaciers" or similar. Transport by icebergs makes me think of iceberg drift after calving.

We are concerned that "sediment transport by tidewater glaciers" would also include transport by the subglacial meltwater system and till deformation, neither of which we consider here.

Line 47: "Gt of sediment per year" – consider explicitly stating that this is sediment from IRD and meltwater, if you have spare words.

Updated as per another reviewer's comment.

Line 57: Sorry, but I don't quite follow how the preceding introduction of Andresen et al. and Overeem et al. points to large uncertainties in the contribution of sediment transport by IRD. As mentioned in the revised text, the given numbers do have large enough uncertainties that they could be reconciled, so I wonder if it would help readers if you were to open the door to IRD as a potentially significant contributor earlier in the paragraph. Would be it correct to add a caveat on line 50 to introduce the idea that, despite the relationship between runoff and core accumulation rates, IRD must still be a significant source? Or provide some nuance about the spatial distribution of the cores vs expected IRD locations?

This statement has been updated to re-emphasize that IRD remains poorly constrained in both prior studies.

Lines 107:110: in my previous review of this manuscript, I questioned the use of a fixed value for sediment concentration in the model given this statement about highly variable sediment concentrations being important for modelling sediment flux. I appreciated your response explaining the cause of this variation between samples and I wonder if some of that reasoning should be transferred to this section of the manuscript, so as not to cause similar confusion or scepticism for readers. Essentially, I think the choice of a fixed sediment concentration is a simplification of reality that is required to make the problem computationally tractable, but, as written, you are making this choice seem scientifically unjustifiable.

This line has been updated to clarify that highly variable sediment concentrations are only found between different basal ice facies.

Line 150: Can you describe how you defined steady-state? E.g. monthly fluctuations of less than x?

Done.

Line 153: I think it would be better if you named the glaciers here, rather than using “southwest” etc, otherwise you imply that they are representative of all of southwest Greenland etc.

Done.

Line 154-156: re time to reach steady-state - I think this would be better placed at the end of the previous paragraph.

Done.

190-192: I wonder if it would be clearer to remove this text. Given the results and conclusions focus on marine terminating glaciers, it’s a little confusing to discuss what are now land-terminating outlets. It’s also not clear what you mean by a “less efficient” catchment.

The text has been edited. We think this is still an important point to include, because it helps explain the regional differences in our model’s predictions of IRD export.

Line 214-215: consider providing again your estimate.

Done.

Line 231: this appears to contradict the statement on line 214/215 because it’s not immediately clear that you’re no longer discussing Greenland’s overall IRD flux. I think it would be helpful if you specified which aspects of the model results you are referring to in the opening sentence.

Clarified in the text.

Line 297-299: I think it would be more accurate to say that “we have observed an increase in mass loss from Greenland’s tidewater glaciers...” because discharge hasn’t increased at the ice sheet-scale since 2005. If you’re tight for words, you could (and maybe should) not even mention mass loss and just go straight to ice discharge, which would let you remove the clause after the comma... Then also modify the following sentence about sediment yield increases accordingly.

Good catch, thank you. The text has been updated accordingly.

Line 403 and elsewhere: you state single parameter values but you tested a range of values. I think you should provide the range here as well as in Extended Data Table 2.

These have been updated where appropriate.

Figure 3: I think the panel labels (a,b) may be missing, or I can’t see them. I think the top panel showing dispersed layer thickness should use the same colormap as the bottom panel – it looks like

they are on the same scale after all and it's odd to use a diverging colormap for non-diverging data. As with Figure 5, it would be great to include the uncertainties as well as the central value.

This has been updated, with the exception of uncertainty values. There was not enough space on the figure to legibly include uncertainty on a catchment-by-catchment basis.

Figure 5: this is a really effective figure! My only suggestions are to (1) move the bold text in the barplot from the top of the error bars to the top of the shaded bar; (2) add lines delineating the ocean sink regions used, and; (3) consider adding uncertainties to the boxes for the ice sheet sectors and ocean regions.

(1) The barplot has been updated. (2) The ocean sink regions are approximate, from Overeem et al. (2017), which has now been noted in the figure caption. (3) Same as above, adding ~ 30 +/- values to this figure drastically reduced the clarity.

*I would like to thank the authors for their thorough revision of the manuscript **Sediment transport by Greenland's icebergs**. The revised version represents improvement in clarity and transparency and has in my view now reached a level that makes it suitable for publication. I commend the authors for their careful and constructive revision of a challenging and significant topic.*

I still have a very few key suggestions for clarification, outlined below.

Reviewer #1

To quantify iceberg rafted debris in Greenland's total sediment budget, Pierce et al combine field data from 210 iceberg samples across three fjords with a numerical model of erosion and entrainment to estimate IRD fluxes. They find that sediment concentrations span a wide range (0.1–45% by mass), and discuss this in context of frozen fringe sediment and dispersed basal sediment. They find a first-order relationship between ice and sediment yield and estimate that icebergs export ~416 Mt/a of sediment, or roughly one-third of Greenland's total sediment flux. Interestingly, the estimated iceberg sediment flux presented in this manuscript is significantly higher than previous estimates reported by Overeem et al., 2017 and exceeds the values anticipated in the discussions in Andresen et al., 2024. However, I have concerns regarding both the field observations used to constrain the model and the approach taken to validate this elevated flux estimate.

Understood. We will attempt to address these concerns below.

Field observations: Hasholt et al, 2022 describes a method to collect samples of ice from icebergs systematically. They point to their 97% uncertainty range based on only 12 icebergs – and their method is meant for future sampling to provide more realistic values as data collection hopefully increases. They write: We present an estimate of the transport related to calving of ~100 million tons per year with a range from 0.3 to 200 million tons. We are fully aware that 12 samples of icebergs are not sufficient to constrain the transport of sediment reasonably accurate, but we have performed the calculation to illustrate the effects of biased sampling. The simultaneous meltwater-driven suspended sediment fluvial contribution is 21–30 million tons (from table 1 in Overeem et al. 2017). The latter number refers to the meltwater sediment flux from the entire Scoresbysund region. For comparison Overeem et al. 2017 find that iceberg sediment accounts to c. 1% of that contributed from melt water (their table 1) for entire Greenland. While I acknowledge the considerable effort involved in collecting and analyzing data from 210 icebergs, I am not convinced that increasing the sample size alone provides greater confidence in the upscaled estimate. Given the inherent variability in iceberg sediment content and the challenges in achieving unbiased sampling, I believe the current dataset is insufficient to support a robust extrapolation to the ice sheet scale. This is not to discount the value of the fieldwork, which is clearly substantial, but rather to caution that the upscaling appears premature.

These statements from Hasholt et al. (2022) are the driving force behind our entire approach in this manuscript. In that work, the authors engaged in a semi-random sampling campaign, choosing the iceberg “closest to the sample position pre-selected on a map.” Then, “the first sample was taken at a random location where the boat bumped up to the iceberg,” with subsequent samples taken approximately 2-6 m from that initial position. This approach led to considerable variability in the qualitative character of the icebergs sampled: they did not necessarily feature debris-rich basal ice layers, or if they did, the repeat samples were not guaranteed to come from those layers. Their 97% uncertainty range includes clean, englacial ice, dispersed ice with relatively low sediment, and solid basal ice with very high sediment load. We agree with the assessment that upscaling such a diverse dataset would be premature.

Ok, great.

However, our strategy here is different, both in respect to how we gathered field data and our process-based approach to upscaling the results. First, we specifically sampled debris-rich basal ice in this study. We included the data from Hasholt et al. (2022) where field notes, photographs, and sediment analyses could reasonably confirm that the samples were obtained from a debris-rich layer of the iceberg. Therefore, our ice-rafted sediment concentrations *are not* representative of a random iceberg (i.e., we do not measure an “englacial sediment concentration”). They are, however, representative of debris-rich basal ice layers. We are confident that our sampled sediment concentrations from icebergs are well within the range of values observed in situ at glacier margins and in laboratory settings (e.g., Dowdeswell, 1986; Dowdeswell and Dowdeswell, 1989; Andrews et al, 1994; Hubbard et al., 1996 and 2009; Cook et al., 2011; Pierce et al., 2024). Yet, it would not be correct to upscale these sediment concentrations by multiplying against total ice discharge, since the vast majority of the ice column is clean ice, not debris-rich ice.

This leads us to a process-based upscaling strategy. The numerical modeling used in this manuscript follows directly from previous theoretical and experimental work on the thickness and extent of debris-rich basal ice layers. Our goal is to predict the thickness of sediment-rich basal ice layers, such that the total sediment flux can be modeled as the product of thickness, concentration, and sliding velocity across the terminus. As such, the iceberg samples we gathered here are only intended be representative of the modeled basal ice layer, and not of any randomly selected iceberg. For further explanation, Pierce et al. (2024) includes a more thorough discussion of these basal ice facies.

Got it. Thanks for the clarification added in line 95 and 96.

We hope that this discussion also clarified the confusion around Overeem et al. (2017). The reviewer points to Overeem (2017) Table 1, however this refers to ‘englacial’ sediment, defined as sediment within the entire ice thickness. Englacial sediment derives mostly from wind-deposited dust, and concentrations in ice cores are observed to be extremely low (0.001%). However, to estimate glacier basal layer sediment flux in that analysis, the authors assumed a Greenland-wide average calving cliff height (300 m), constant debris layer thickness (3 m), and a sediment concentration of 20% by volume (from sparse field data by Dowdeswell, 1989 and Andrews, 1994), or roughly 50% by mass. In table 1 from Overeem et al. (2017), the 1% value refers to the volume flux of basal ice relative to total calving ice. The basal sediment flux is estimated as 1.92 Gt (Table 1, column 3). Hasholt et al. (2006) estimated a sediment flux associated with Greenland Iceberg calving of 50-500 Mt/yr, based on extremely sparse data. Our model vastly improves on the previous estimate by simulating the processes that form the basal ice layer, rather than making any assumption about its size relative to the calving front. Our current paper also demonstrates that sediment concentration in the basal layer consists of high concentration frozen fringe and low concentration dispersed sediment (while appearing visually dirty throughout the full basal layer in field views of flipped icebergs). Consequently, our process-based approach sharpens previous estimates, while revising the total amounts substantially downward.

Andrews, J. T., Milliman, J. D., Jennings, A. E., Rynes, N. & Dwyer, J. (1994). Sediment thicknesses and Holocene glacial marine sedimentation rates in three East Greenland Fjords (ca. 68° N). *J. Geol.* 102, 669–683.

Cook, S. J., Swift, D. A., Graham, D. J., & Midgley, N. G. (2011). Origin and significance of ‘dispersed facies’ basal ice: Svinafellsjokull, Iceland. *Journal of Glaciology*, 57(204), 710-720.

Dowdeswell J.A.(1986) The Distribution and Character of Sediments in a Tidewater Glacier, Southern Baffin Island, N.W.T., Canada, *Arctic and Alpine Research*, 18:1,45-56

Dowdeswell, J.A., Dowdeswell E.K. (1989). Debris in Icebergs and Rates of Glaci-Marine Sedimentation: Observations from Spitsbergen and a Simple Model. *The Journal of Geology* 97, 2, 221-231.

Hasholt, B, Bobrovitskaya, N., Boge, J., McNamara, J., Mernild, S., Milburn, D., and Walling D. (2006). Sediment transport to the Arctic Ocean and adjoining cold oceans, *Nordic Hydrology* Vol 37 No 4–5 pp 413–432

Hubbard, B., Sharp, M., & Lawson, W. J. (1996). On the sedimentological character of Alpine basal ice facies. *Annals of Glaciology*, 22, 187-193.

Hubbard, B., Cook, S., & Coulson, H. (2009). Basal ice facies: a review and unifying approach. *Quaternary Science Reviews*, 28(19-20), 1956-1969.

Pierce, E., Overeem, I., & Jouvett, G. (2024). Modeling sediment fluxes from debris-rich basal ice layers. *Journal of Geophysical Research: Earth Surface*, 129(10), e2024JF007665.

*I believe Overeem et al.'s Table 1, which I consulted when reviewing this manuscript, contains inconsistencies in the reported iceberg-rafted sediment flux. First, the **basal ice calving flux** is listed as 1.92 Gt yr^{-1} , whereas both the methods and main text specify 2.88 Gt yr^{-1} . Second, in the lower part of the table, the final **Greenland calving flux** is given as 0.014 Gt yr^{-1} , representing only the englacial component and omitting the basal debris contribution. It is unclear why this basal component was excluded, given that it accounts for roughly 3 of the total 9–12 Gt yr^{-1} , i.e. about one-third to one-fourth of Greenland's sediment export. However, I assume the table was designed to compare suspended sediment fluxes from Greenland with global fluvial sediment discharge. In that context, it makes sense that the authors only included meltwater-borne and englacial fine sediment, since these are directly comparable to riverine suspended loads. The basal debris-rich ice flux, though substantial, represents a different transport mechanism (sediment carried in solid ice rather than in suspension), and is based on assumed rather than observed parameters. However, the way the table is presented is somewhat confusing, as the omission of the basal component is not explicitly stated and the values appear to represent **Greenland's total calving-derived sediment flux**. This is a highly cited paper, so it would have been helpful to emphasize that Greenland's total sediment export is higher when the basal contribution is included. In the current manuscript (line 71), the 1.92 Gt yr^{-1} value is referenced, but the methods section in Overeem et al. reports 2.88 Gt yr^{-1} . The current study certainly doesn't stand or fall on this point, but it is confusing when table values do not match the referenced calculations.*

However, with the new study, values of IRD flux are effectively put back in place, further justifying the importance and publication of this manuscript.

See also my later comment on summarizing the current state of knowledge on Greenland's sediment budget - specifically, the existing quantifications and which melt components and glacier types each study includes.

As a first step, a more informative approach would have included a detailed characterization of the sediment contained within the sampled icebergs—particularly grain size distributions and lithologies—as well as descriptions of sediment banding features, including band thickness, contact relationships, and three-dimensional morphology. This is only partly touched upon – but a systematic report and analysis of the iceberg data from the 210 bergs, including photographs, would be very valuable for our understanding of Greenland's 'muddy bergs'. Greenland's iceberg population is indeed highly heterogeneous, and a deeper understanding of this variability is needed before attempting large-scale quantification of IRD fluxes.

We agree that there is considerable variability in Greenland's icebergs, and that variability is inherently worthy of further study. We do not agree with the assessment that the rest of the work presented in this manuscript is unworthy of attention until such time as the variability in field observations can be addressed. There are many studies cited throughout the manuscript that use field observations, laboratory experiments, and theory to constrain the processes responsible for building debris-rich basal ice layers. These studies generally agree on a range of sediment concentrations and layer thicknesses resulting from basal sediment entrainment – although again, we agree there is further work to be done. However, modeling a bulk sediment flux does not rely on a lot of the complexity noted here; properties such as grain size distribution, lithology, banding

features, etc. are important for understanding the underlying processes at play, but do not directly control the overall sediment flux.

Validation: The only validation provided in the manuscript is a comparison with sediment flux estimates from Overeem et al., 2017 and Andresen et al., 2024. Pierce et al. argue that the difference between the two studies reveals an unaccounted sediment fraction attributable to icebergs, noting that Overeem reports suspended sediment flux, whereas Andresen reports total sediment flux. However, this interpretation is not correct. Andresen et al. estimate total sediment flux only from marine-terminating glaciers at 1.324 ± 0.791 Gt/yr, while Overeem et al. report two estimates of suspended sediment flux from both marine- and land-terminating glaciers: 0.892 Gt/yr (erosion-based) and 1.28 Gt/yr (meltwater discharge method) (and both studies btw point to a low contribution from iceberg-derived sediment). So, the difference between the two studies clearly cannot be used as a proxy for iceberg sediment flux. A short summary of the two papers, for the readers not deeply acquainted with these studies, would have clarified this.

We appreciate the points raised here, as it appears our exposition of both papers in question was lacking. We agree with this summary of Andresen et al. (2024), although we do not think that the results in that study preclude a larger contribution from iceberg-rafted sediment. In the Sermilik fjord diamicton, they note that the sand content is roughly 20% in cores, whereas sand content in basal ice layers is 50-80%. This still allows for 25-40% of the sediment to be delivered by icebergs, comparable to what we predict here. Of course, this analysis relies on major assumptions about the variability and preservation of grain size distributions in transit. In the plumite, sediment > 63 microns only composes $\sim 1\%$ of the total accumulation. Note, however, that this is not necessarily inconsistent with our approach. We estimate the total sediment contribution from IRD as approximately $1/3$ of the sediment reaching fjords, but that is not likely to be true on a fjord-byfjord basis. Sermilik fjord's outlet glaciers have a moderate ice yield (due to very large catchment areas) and high runoff rates (as is typical of southeast glaciers). So, it is entirely reasonable that the IRD fraction is much lower than the meltwater fraction in this setting. Also, as they note, the Andresen et al. study does not have access to cores within appx. 30 km of the calving front. Given that the basal ice layers with the highest sediment concentrations are located near the bed, we would expect those layers to melt out earliest, and thus closest to the calving front. We agree with this review that this is an important discrepancy to address directly in our manuscript.

Please see my comments further down regarding the use of IRD deposits from marine sediments to compare with your model results.

Regarding Overeem et al. (2017), there may be additional clarifications required. First, the upscaling approach did focus on meltwater-driven sediment transport in both marine-terminating and landterminating glaciers, as stated here. But, it is important to note that sediment from land-terminating glaciers ends up in proglacial rivers, which drain into fjords. Thus, the vast majority of sediment from these two pathways of transport (meltwater and icebergs) end up at the same sink. Further, Overeem et al. (2017) do provide two different estimates of suspended sediment flux, but actually point to a massive contribution from icebergs of 1.92 Gt per year (as opposed to the relatively low contribution of ~ 14 Mt/yr from englacial sediment). We describe the estimation methodology above, but clearly the error in that approach lay in assuming that the entire debris-rich ice layer of 3 meters had a uniform sediment concentration around 50% by mass. The current manuscript revises this estimate down by more than half, based on a much more thorough understanding of the typical ranges of sediment content in basal ice layers.

The motivation and approach for the current study is well laid out in the manuscript.

Despite these clarifications, we do understand the overall point here: it is not fair to use the difference between estimates in Andresen et al. (2024) and Overeem et al. (2017) as a validation step. We have clarified throughout the manuscript that this is a discussion point – that is, a potential consequence of our approach, rather than a piece of evidence in its favor. We have also added a paragraph discussing more local validation steps where possible.

I appreciate that the revised manuscript now treats the comparison between Overeem et al. (2017) and Andresen et al. (2024) as contextual discussion rather than as a validation test. This change addresses the main issue I raised previously. Overeem et al. report suspended sediment fluxes from both land- and marine-terminating glaciers (but actually do not include basal IRD in their final quantification, despite calculating high numbers), whereas Andresen et al. quantify total sediment fluxes (melt plus IRD) only from marine-terminating glaciers. These differences in glacier type and process coverage mean that the two datasets cannot be used to infer a residual iceberg-derived component. The revised manuscript now states that the two studies “could be reconciled into a consistent sediment budget” and that “both point to large uncertainties in the contribution of sediment transport by IRD” (lines 55–57). While this is a more balanced framing, I suggest clarifying around line 47, to avoid confusion, that Andresen et al. (2024) estimate sediment export only from marine-terminating glaciers to fjords, to make the scope distinction explicit and prevent misinterpretation of the comparison (or those comparisons coming later).

Line 44-50: The authors specifically write: “Although IRD contributes sediment to the North Atlantic Ocean, Greenland’s fjords are the primary sink for sediment sourced from beneath the ice sheet. In a recent study, Andresen et al. estimated that Greenland’s fjords receive 1.324 ± 0.79 Gt of sediment per year (appx. 4.9×10^8 m³) [18]. This study identified a relationship between accumulation rates in marine sediment cores and meltwater runoff from adjacent catchments, suggesting that fluvial sediment transport plays a key role in delivering material to Greenland’s fjords.” The correct phrasing here would be: “Although IRD contributes sediment to the North Atlantic Ocean, Greenland’s fjords are the primary sink for sediment sourced from beneath the ice sheet. In a recent study, Andresen et al. estimated that Greenland’s fjords receive 1.324 ± 0.79 Gt of sediment per year from all of Greenland’s marine-terminating glaciers (appx. 4.9×10^8 m³) [18]. This study identified a relationship between accumulation rates in marine sediment cores and meltwater runoff from adjacent catchments, suggesting that subglacial sediment transport plays a key role in delivering material to Greenland’s fjords.”

Line 205-208: the authors write: “Despite these caveats, sediment fluxes from IRD and meltwater fall within the range of uncertainty of fjord accumulation rates, suggesting that these are the two most significant pathways of sediment transport from the Greenland ice sheet.” I am not sure what other significant pathways of sediment transport there are than icebergs and meltwater.

With Overeem et al. (2017) estimating total sediment flux from both land- and marine-terminating glaciers (excluding basal IRD), Andresen et al. (2024) combining meltwater and IRD fluxes from only marine-terminating glaciers, and now Pierce et al. focusing only on IRD, the community has three complementary and independently validated estimates of Greenland’s sediment export. Why not summarize this in the manuscript? It is entirely appropriate to compare these studies to assess how their results overlap, and a clear outline can avoid confusion about which flux components are included in each estimate.

Moreover, the methodologies in these studies differ significantly. Andresen et al. rely on sediment core accumulation rates, while Overeem et al. use remote sensing and modeling of meltwater fluxes. Both approaches involve large uncertainties, which are acknowledged in their respective discussions in these papers. However, the broad agreement in their flux estimates, despite such different methodologies, suggest they are both on the right track. But their differences reflect these methodological differences and limitations—not a missing ~ 430 Mt/yr of sediment. For example, Overeem’s method is limited by sparse data near deep glacier termini, which alone could account for substantial underestimation. Likewise, Andresen study is

limited by not having sediment cores right next to the glacier margins.

The manuscript's interpretation of this discrepancy as iceberg-derived sediment is therefore problematic, especially as it forms the sole basis for validating the model's sediment yield.

It is true that the methodologies in both studies differ, but it is also important to note that the studies are not measuring the same components of the system. Andresen et al. use sediment core accumulation rates, so their upscaled result should be viewed as the total sum of sediment that reaches fjords. (*← only marineterminating*). Overeem et al. use a remote sensing method to connect meltwater fluxes with suspended sediment load, and thus are only able to predict the contribution from suspended sediment (not bedload, nor iceberg sediment). It would be odd if the studies were in close agreement, since sediment does not exclusively reach fjords through suspension in meltwater. Note that we do not include Overeem et al.'s assessment of ice-rafted debris fluxes or englacial sediment fluxes here. We do elect to use the upscaling relationship between glacial erosion and sediment flux from Overeem et al. with a stronger fit to the field data, as that was the value the authors themselves chose to elevate throughout the manuscript, but we made that decision more transparent in the current manuscript.

Yes, and this is clear. I also refer to my point above.

A question that comes to mind here is: Why have they not compared with sediment core data from Greenland's fjords and shelf presenting iceberg rafted debris fluxes?

We feel this comparison is out of scope of this study. There is an inherent problem in using sediment core data to validate bulk fluxes, as noted by the discussion brought up earlier from Andresen et al. (2024). Sedimentologists identify IRD as coarse, angular sediment grains in marine cores that would have fallen out of suspension in a plume much closer to the terminus. However, a substantial proportion of IRD derived sediment (anywhere from 20-50%, depending on the study) is fine enough to be indecipherable from fluvial suspended sediment (f.e. Smith and Andrews, 2000). And, to further complicate matters, grains may be worked by the subglacial fluvial system before becoming entrained in basal ice, such that angularity is not a precise identifier either. Most studies of IRD in cores use sedimentological evidence – e.g., magnetic susceptibility – to identify phases of elevated/diminished IRD fluxes. However, this does not always translate neatly back into a flux in mass/time or even mass/area/time (the second of which has an inherent problem with deposition location – see the points about Andresen et al., 2024, above). Despite these reservations, we have added a comparison to a few different validation approaches that may be appropriate for this work. Smith, L. M., & Andrews, J. T. (2000). Sediment characteristics in iceberg dominated fjords, Kangerlussuaq region, East Greenland. *Sedimentary Geology*, 130(1-2), 11-25.

Most studies of IRD around Greenland use sand flux as a proxy for IRD, rather than characteristics such as grain angularity or magnetic susceptibility. I commend the authors for taking the opportunity to compare their results with such fjord-based IRD studies (i.e. sand flux) in the revised manuscript (lines 249–259). This comparison is well within scope, especially since the authors highlight the relevance of IRD in paleoclimate research. While discrepancies between studies are to be expected, reporting such observations is essential for advancing our understanding of key processes - both beneath the ice sheet (fringe formation and regelation) and within fjord waters. I find the discussion very interesting and motivating. In line 260 the authors write: "... to help validate (or complicate) our understanding of IRD fluxes from frozen fringe". I suggest writing "to improve understanding of the mechanisms involved in IRD melt from frozen fringe"

Line 71: the authors write: "Not only is this estimate high compared to reconstructed accumulation rates in fjords...etc" – a reference is needed for this statement.

The authors also write (Line 297-299): We are observing increased mass loss from Greenland's tidewater glaciers as ocean warming contributes to enhanced rates of iceberg calving, increasing the ice sheet's overall ice yield [39-41]. There is a wealth of paleoclimate/IRD studies that shows this. One examples amongst many is: Jennings et al. 2017: Ocean forcing of Ice Sheet retreat in central west Greenland from LGM to the early Holocene - ScienceDirect

Some minor comments

Line 33 and 34: Some of these studies also point to plume sediment as a shoal-builder, in particular the studies of temperate Alaskan glaciers.

Good point. The sentence is now: "Within the grounding zone of glacial outlets, sediment deposits contribute to shoal formation, inhibiting tidewater glacier retreat."

Line 36: heightened levels of chlorophyll are linked with sediment in the Southern Ocean where iron is a limited nutrient. Around Greenlandic icebergs local upwelling of macronutrients from Atlantic-sourced water may be the cause of heightened chlorophyll, and not the elements contained in the sediment. So, it would be correct to mention these observations goes for the Southern Ocean.

Thank you for identifying the flawed statement here. We have updated it in the revised draft.

Line 37 and 38: The authors write "While upwelling of deep ocean water likely supplies more nutrients to Greenland's coastal oceans than direct inputs from the ice sheet, icebergs are still an important source of micronutrients, such as iron and manganese" – I suggest: "While upwelling of deep ocean water is considered an important supplier of macronutrients nutrients to Greenland's coastal oceans, direct inputs from the ice sheet icebergs are an important source of micronutrients, such as iron and manganese".

Line 113: what are the characteristic sediment grain size distribution in the frozen fringe sediment, and in the dispersed basal ice facies according to the referenced literature? And are these based on in situ observations by landterminating margins? This information is highly relevant when linking the sediment sampled from icebergs with the sediment entrained underneath the glacier. Do the grain size distributions match?

We only have limited quantitative grain size analysis for most of the IRD samples, and none from most of the till that underlies the Greenland ice sheet. *Ok. For future work, it would be interesting with grain size distributions from the iceberg IRD samples to be compared with for example those in Baltrūnas et al. The sedimentology of debris within basal ice, the source of material for the formation of lodgement till: an example from the Russell Glacier, West Greenland. *Geologija* 51, 12-22 (2009).*

The grain size distribution in the frozen fringe is inherited directly from the grain size distribution in the till beneath the glacier (this is a fundamental part of the theory on frozen fringe mechanics). Particle migration into dispersed basal ice, above the frozen fringe, is an open question in glaciology. We have used the model from Pierce et al. (2024) in this work, as we believe that model represents the highest fidelity to real-world observations from glaciated margins.

Line 154. What is the role of subglacial meltwater derived from surface melt? Does channelization of meltwater near the glacier margin influence the sediment content at the icebed interface? This likely represents an important process through which basal sediment is transferred into the meltwater plume. If this mechanism is not accounted for, estimates of sediment yield from icebergs may be significantly overestimated. While these processes are reported on in the methodology section is not mentioned in the discussion and it is not clear to me how IRD and plume sediment is partitioned in this study.

We agree that subglacial fluvial transport is likely a dominant sediment source, especially in many of

the outlets in southwest Greenland. We do not explicitly simulate subglacial fluvial transport here, nor do we simulate meltwater plumes. This model is focused on basal sediment entrainment, although we do agree that the interactions with the fluvial system are certainly interesting and worthy of further study. We make the case here that frozen fringe is active beneath a vastly greater area of the ice sheet than channelized meltwater, such that these interactions would be limited to a relatively small area of the bed. Importantly, there is sufficient erosion in our model to allow for sediment transport by both pathways – that is, we would not expect our IRD fluxes to be greatly reduced by adding a fluvial component to the model. However, modeling subglacial sediment transport by meltwater is itself an open question in glaciology, such that a fully coupled model of the system would be a vision, but far out of scope for this work.

Ok. I also note that these considerations are already well described in the methods section.

Line 156-157: How do the erosion rates compare to previous reported estimates of erosion rates from Greenland catchments? I believe the ones reported by Pierce et al. are 100-fold higher than previous estimates?

Good catch. The x-axis label was incorrect on this plot. We have elected to remove it, as it raised significant confusion among all reviewers and did not garner much positive feedback. Our peak erosion rates are that high, but only at limited points in space. The catchment-averaged erosion rates are on the order of 0.01-0.1 mm per year.

Line 165: is this not expected when the same surface velocity and terminus width are primary variables in calculations of both sediment yield and ice yield (data in Mankoff)? The derivation of ice yield is reporting in the methodology section, but there is no reference to this on the Figure 4C.

In our understanding, an appropriate null hypothesis would be that sediment discharge scales with ice discharge (e.g., Methods section of Overeem et al., 2017). And as pointed out here, that is precisely because surface velocity and terminus width are used in both calculations. But, that is not what we find here. Instead, we find that the ice yield (surface velocity x terminus width x calving front height) scales the sediment yield (sliding velocity x terminus width x debris-rich ice thickness). While surface and sliding velocity are related, they are not equal to one another for many of these systems. And the thickness of debris-rich ice layers (our primary model output) is fully independent of any other variables used in the ice discharge or yield calculation.

Line 245-248: I suggest to delete this statement. Sea level will decrease by Greenland's glacier due to isostatic rebound and reduced gravitational pull; it will not increase here. The question is if shoaling can prevent incoming warm Atlantic waters and 50 cm of sediment per decade would do nothing here.

Good catch, thank you for pointing out the flawed logic. We have exchanged that sentence for a comparison with a grounding zone wedge identified at Whillans Ice Stream.

Line 228: grain sizes and angularity....would be great to be more specific here.

Added more specific descriptions.

Line 230: how is that known?

We are not sure what this statement is referring to. We know that prior to iceberg calving, dispersed basal ice melts out later than frozen fringe because it is located higher in the ice column. While calving and subsequent mixing in melange can stochastically alter iceberg orientation, it is very unlikely to find a geometry where dispersed basal ice melts out earlier than the clean ice or frozen

fringe on either side of it.

I was referring to the statement: “The dispersed layer also contributes to the total amount of IRD exported in a catchment, but is dominated by fine sediment with a much lower proportion of large particles.” How do the authors know this, given that grain size composition was not measured? A reference to theoretical or observational work supporting this claim would strengthen the statement.

Line 232-234: to be clear to a broader audience – specify what is meant by reconstructions. But be also aware that reconstructions of iceberg production and dispersal changes in the past are based on the coarse fraction (for the reasons mentioned by Pierce et al.).

Clarified further in the text.

Line 235-237: I agree, and for this reason grain size distributions of the samples would have been informative.

Unfortunately, we do not have quantitative grain size distribution or chemistry measurements for all of the IRD samples, which is why we omitted this characteristic from the discussions. This is a good point, and will have to wait for future work. For the current paper, we chose to focus on the physical properties that are most influential for bulk sediment fluxes.

Ok

Line 250-252: it says: “ This assumption holds if the system is limited by supply- that is, if the glacier has the capacity to transport all available sediment.” Should it not be “is not limited by supply”?

Apologies for the confusing wording here. If the system is limited by supply, this implies that all available sediment is moved. Otherwise, it would be limited by the glacier’s ability to transport the sediment. Regardless, this paragraph is not a critical part of the discussion, and has been removed to make space for more relevant topics raised earlier in this review.

ok

Reviewed by Camilla Andresen